# Seafaring and navigation in the Nordic Bronze Age: The application of an ocean voyage tool and boat performance data for comparing direct open water crossings with sheltered coastal routes

**Boel Bengtsson**[1]*, **Alvaro Montenegro**[2], **Ashely Green**[1], **Matteo Tomasini**[3], **Martyn Prince**[4], **Victor Wåhlstrand Skärström**[3,5], **Knut Ivar Austvoll**[6], **Johan Ling**[1], **Cecilia Lindhé**[3]

**1** Department of Historical Studies, University of Gothenburg, Gothenburg, Sweden, **2** Department of Geography, Ohio State University, Columbus, United States of America, **3** Gothenburg Research Infrastructure in Digital Humanities, University of Gothenburg, Gothenburg, Sweden, **4** Wolfson Unit, University of Southampton, Southampton, England, **5** Signal Processing and Biomedical Engineering. Chalmers University of Technology, Gothenburg, Sweden, **6** Department of Archaeology, Conservation and History, University of Oslo, Oslo, Norway

* bessemer.clark.boel@gu.se

## Abstract

This study presents an "ocean voyaging tool" that combines predicted vessel performance data with agent-based simulations. This tool offers a new way to assess navigation and seafaring abilities in prehistory while also enabling the direct comparison between different type vessels and vessel configuration, navigational skills and propulsion. Results are filtered using certain limitations on safety (wind strength, wave height, light etc.) and navigational error. The method is here used to compare direct open water (c. 110 km) and coast hugging (c. 700 km) voyages between Jutland and south-west Norway in the Early Nordic Bronze Age, two areas that were closely connected from the Late Neolithic throughout the Bronze Age (c. 2350-1500 BC). Simulated results suggest that although the longer coastal route is usable all year round, direct open sea voyages, which included navigation out-of-sight-of-land for up to 50 km, were most likely undertaken. Such voyages would have necessitated boats capable of withstanding and maintaining directional control in ≤ 1 m waves and winds of up to 10 knots (5 m/s) at a minimum. Furthermore, these simulations highlight the comparable advantage of sail over paddling for transporting cargo over long distances (journeys of more than one days length).

## Introduction

Boats and maritime technology play a pivotal role in the movement of peoples, ideas, goods and technology in prehistoric societies. This becomes apparent in instances where cultural traits such as metal artefacts, iconography and, increasingly, ancient DNA belonging to the same group of people are found on land on opposite sides of large stretches of open water [1].

**Data availability statement:** The input currents and wave data to run the simulations is obtainable from the Copernicus Marine Service (product DOIs: 10.48670/moi-00013, 10.48670/moi-00014, 10.48670/moi-00059 and 10.48670/moi-00060); the input wind data are obtainable from the Copernicus regional reanalysis for Europe (CERRA) website (DOI: 10.24381/cds.622a565a). The code for the simulations, as well as the aggregate of the results and trajectories are obtainable on Github at https://github.com/mtomasini/PLOS-Voyager.

**Funding:** This work is supported by Riksbankens Jubileumsfond under Grant M21-0018. There was no additional external funding received for this study. The funders had no role in study design, data collection and analysis, decision to publish, or preparation of the manuscript.

**Competing interests:** The authors have declared that no competing interests exist.

For instance, the Nordic Bronze Age societies exchanged metals over great distances with up to one tonne imported from the British Isles alone during a period from around 1600 to 1500 BC, something which would have provided an impetus for the development of boat technologies [2,3]. Recent research stresses the expansive dynamism of the Nordic Bronze Age groups because of their ability to organize, and capitalize on, a complex Maritime Mode of Production (MMP) [2]. The MMP included political strategies to control trading and raiding through owning boats and financing long distance maritime expeditions. These expeditions would actively seek out the sources of, e.g., metals, and thereby effectively cut out any expensive middlemen and lower-level transactions as well as potentially hostile encounters in the process [2,4].

Yet, the boats, the navigational skills and the actual sea routes that undoubtedly underpinned such occurrences remain elusive and the complex systems integrating these activities are most often illustrated merely by the addition of large arrows on a map.

Simulations of seafaring voyages as method to understand maritime mobility in prehistory were first developed in the 1970s with a view to exploring whether the Polynesian expansion could have happened by chance, that is, through passive voyages where boats were simply drifting with winds and currents [5]. More recent simulations focus on passive drift scenarios, sea journeys based on very rough estimates of paddling speed, or the simulation of historical voyages where there are detailed diaries of passage and course [6–13]. Other efforts to isolate some of the multiple variables needed to understand prehistoric seafaring, include the application of least cost models, where winds, currents and topography information is used to generate routes that minimize traveling "costs", such as duration or risk [14, 15].

Meanwhile, there have been enormous improvements in digital hardware and software as well as in the resolution of the wind, currents, and other environmental data simulations are dependent on. For example, there are several ocean voyaging optimization software products on the market developed for use within the commercial shipping industry and amongst the boat racing community (see for example DECARTES & ADRENA). Attempts to apply aspects of such optimization models, using, e.g., weather routing linked to land visibility and the detection of specific landmarks, has also recently been used as a way to explore prehistoric sea routes [16]. However, none of these applications allow for agency, which is seafarers making active choices based on specific conditions and their knowledge of their boat. For this, reliable boat performance data is needed [7].

The aim of this paper is to present a way to use ocean voyage modelling together with high resolution performance data of a reconstruction of the 350 BC Hjortspring boat, here referred to as a Bronze Age (BA) type boat [17], to go beyond large arrows on a map and try to gain an understanding of the nature of prehistoric sea crossings, limiting factors for different types of crossings while considering aspects of safety and seafaring abilities built up over centuries.

As a case study, we are going to simulate and compare sea journeys between the region of Thy in northern Jutland, Denmark, and the region of Lista, in southwestern Norway (Fig 1), two non-tidal areas that are seemingly divided by over 100 kilometres of open Skagerrak waters, yet, for which the archaeological evidence suggest regular and close contact from the Late Neolithic period into the Bronze Age [19].

A direct route between these two regions will be compared to the much longer (c. 700 kilometres) but safer route afforded by using the inland waterway of the Limfjord to the east, crossing the Kattegat via the island of Læsø and from there meeting up with the sheltered waterways, or "highways", offered by the archipelagos of the Swedish west coast and continuing northwards to the Oslo fjord (Fig 1) [20]. From the Oslo fjord area, the onward sea journey follows the coast to the west, using islands, natural bays and harbours for shelter on the final leg to Lista.

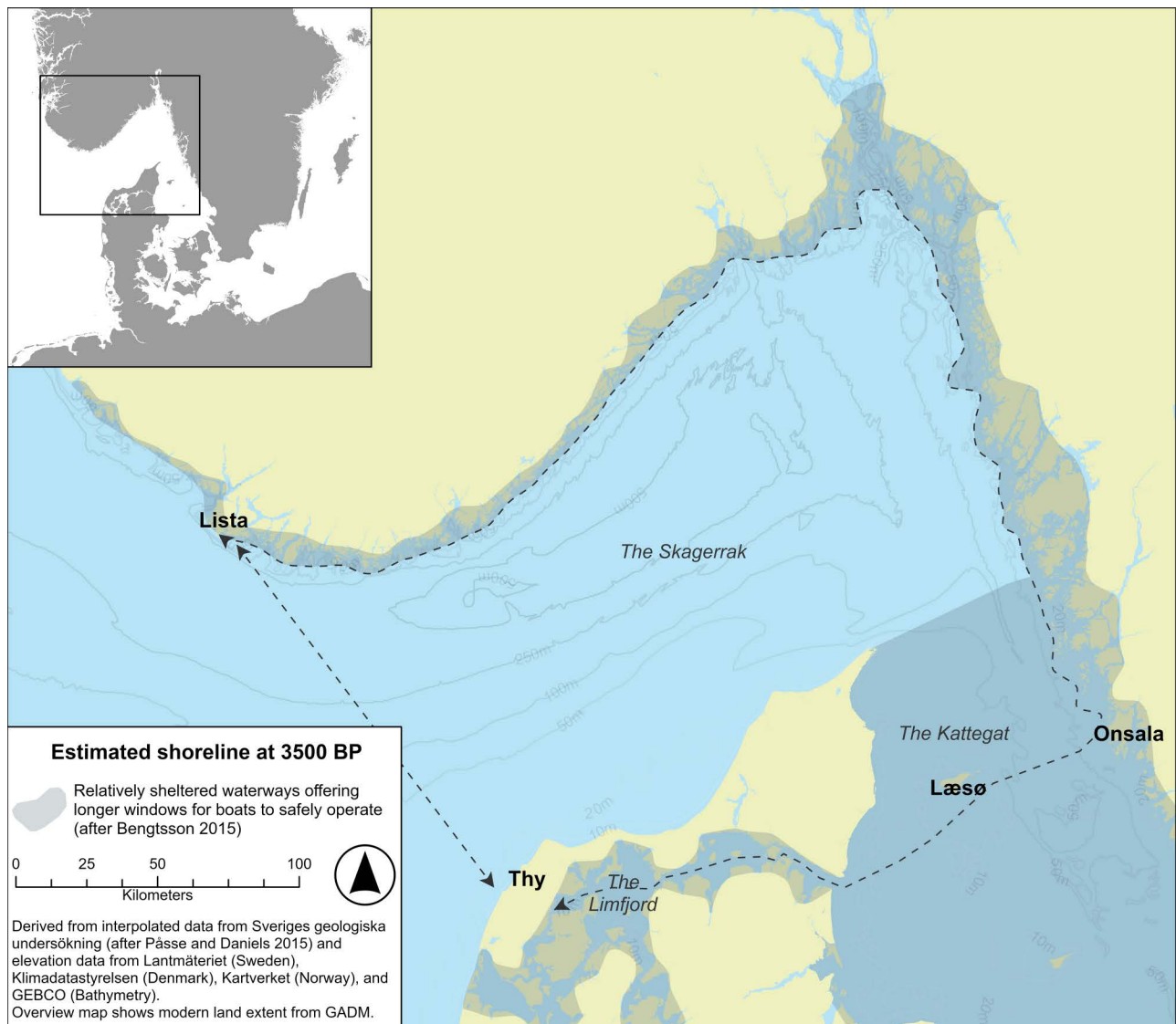

**Fig 1. The geographic locations of Thy and Lista in southern Scandinavia, with the direct and coastal route marked out.** This map shows a paleo-DEM created by the authors in ArcGIS Pro using elevation data from Lantmäteriet, Klimadatastyrelsen, Kartverket, and GEBCO (GEBCO Compilation Group (2024) GEBCO 2024 Grid (https://doi.org/10.5285/1c44ce99-0a0d-5f4f-e063-7086abc0ea0f)) under a CC-BY license and shore displacement values provided by Sveriges geologiska undersökning (SGU) (after [18]Påsse and Daniels 2015). Note: None of the original elevation data is displayed.

The main questions we seek to answer with these simulations are:

(1) What are the limiting factors of a paddled vessel such as the Hjortspring boat for under-taking long distance sea journeys across open sea (wind, sea state, leeway drift, day light, supplies in terms of water, food, clothing, cargo).

(2) How often would or could such sea crossing have taken place and at what times of the year? (Bearing in mind occurrences of, e.g., cloudiness and fog during otherwise, "ideal" conditions of little wind).

(3) How much cargo of goods could be transported between the two regions using a paddled BA type boat?

(4) How do direct journeys over the Skagerrak compare to the longer sea route that follows the sheltered coastline in terms of duration, safety and frequency at which journeys can be completed?

(5) Are there any differences in the duration or in the number of windows of opportunity for direct sea journeys from Thy to Lista in comparison to direct sea journeys in the opposite direction?

## Archaeological background

The area of Thy is situated in northern Jutland, Denmark, on the northern shores of the sheltered inland waterways afforded by the Limfjord and with the exposed sandy seashores of the Skagerrak to the north (Fig 2). The strategic location on the Limfjord, which offered a relatively safe and sheltered east-westerly seafaring route, connecting the North Sea with the Baltic Sea up until its western entrance silted up in the Middle Ages [21], no doubt helped ensure its position as centre of wealth and power from the Late Neolithic period into the Bronze Age [22, 23].

Across the Skagerrak strait, the small peninsula of Lista at the very southern tip of Norway, is recognised by good agricultural land, sandy beaches, smaller inlets, waterways and fjords that could serve as portages in order to avoid the more exposed and dangerous stretches of sea around the peninsula [24]. The peninsula has in historical times been of great strategic importance both as a protective harbour and portage route for sailing ships, but also as a control point during the German occupation during World War II. From prehistory, the region is recognised by a material assembly and architectonic expression that is closely connected with that of Jutland, specifically from the Late Neolithic and onward. This resemblance has been pointed out by several researchers over the years [19,24–28]. Already in 1869 [29], Worsaae pointed to a similar material expression between Jutland and the southwestern part of Norway, and later in 1877 Anders Lorange surveyed many of the Bronze Age mounds at Lista [30]. Above all, it is the large amounts of type I flint daggers, introduced at the beginning of the Late Neolithic that show strong ties with Jutland [31]. An estimated 10 percent of all type I daggers in Scandinavia can be found along the coast of Norway [31]. In part, these daggers along with other beaker elements such as wrist guards, flint sickles, tanged and barbed arrowheads, and small composite mound burials have been connected to groups in the Limfjord region [32, 33]. The emergence of this material assembly is strongly connected to a western oriented coastal route that seems to begin in the Lista region and stretches all along the western coast to Nordland [34]. Following the archaeological objects is also a new socioeconomic shift with the emergence of two-aisled longhouses and agropastoral subsistence economy that would come to define the following Bronze Age [35]. For all this to have been possible a new maritime technology must have been a prerequisite, and several researchers connect this material and socioeconomic change with the first planked-built vessels, the preludes to the Hjortspring boat [19,36,37].

The initial introduction of this socioeconomic package is later followed by the first metal finds in Norway, nearly all with a western oriented distribution [38]. Several of the early metal objects are located on the small peninsula of Lista [39]. This includes a Late Neolithic noppenringe from the continent, an early tin awl and later tin-alloyed bronze objects dated to Early Bronze Age Period I with origins to the continent [4,39]. Metal objects accumulate during the Bronze Age proper, between c. 1500–900 BC, many of which are discovered within monumental earthen barrows and cairns that are clearly visible from the seaway. The constructional choice of burying the dead in earthen barrows (limited to southwestern Norway) is closely tied to South Scandinavian practices and a sign of wealth and power, most likely accumulated due

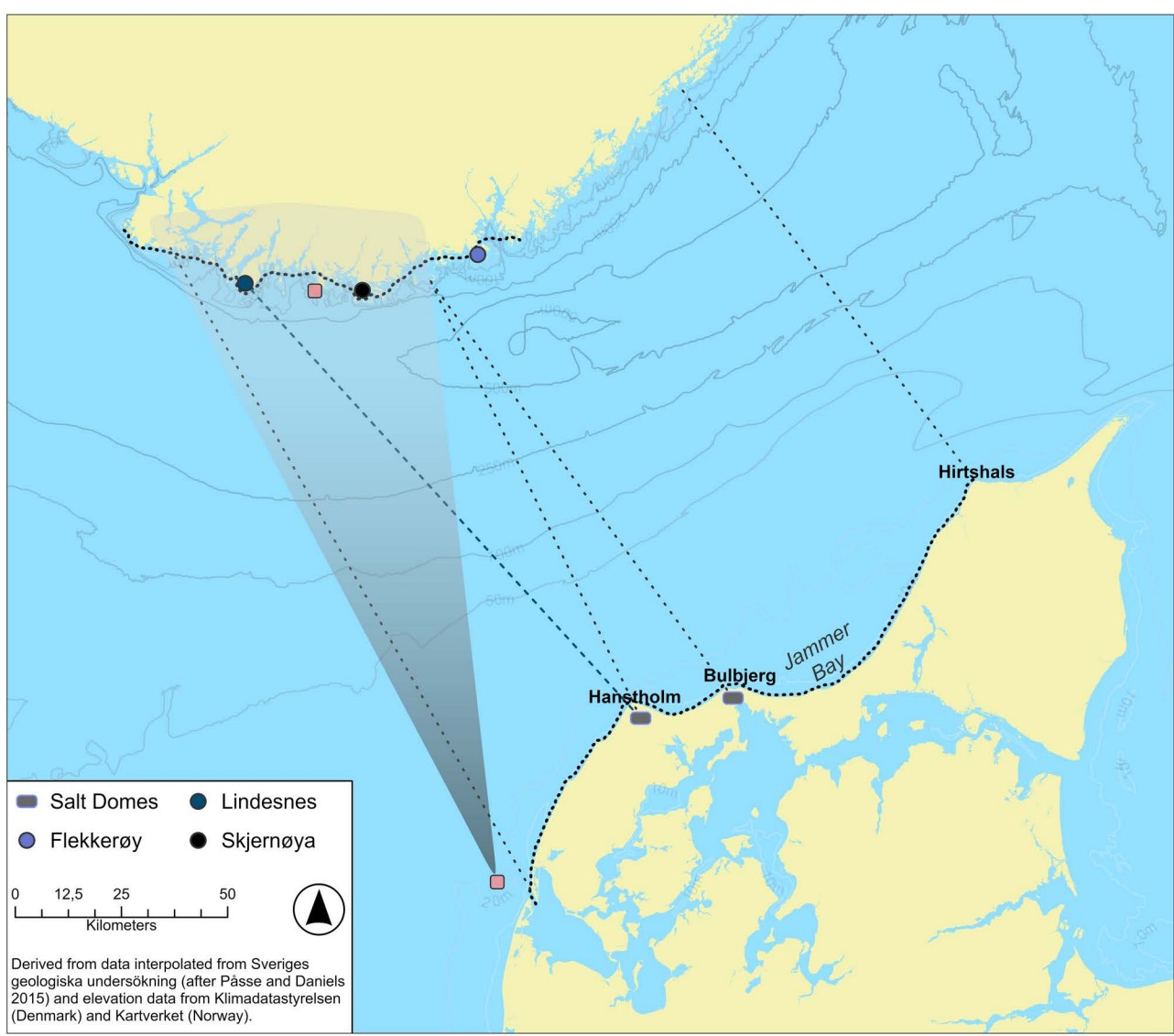

**Fig 2. Routes from Thy to Lista as suggested by various authors (after Johansen 1986, Kvalø 2000, Marstrander 1950 and Østmo 2005), with approximate distances on a c.** 1500 BC paleogeographic map of the area. The red and blue squares indicate the starting/ending points of routes simulated in this paper. The shaded area represents the estimated radius sector of arrival when allowing for the anticipated navigational error on a northbound crossing. This map shows a paleoDEM created by the authors in ArcGIS Pro using elevation data from Klimadatastyrelsen and Kartverket under a CC-BY license, and shore displacement values provided by SGU (after Påsse and Daniels 2015). Note: None of the original elevation data is displayed.

to Lista's strategic location as a bottleneck ([40] for a discussion on the concept). As emphasised by Prescott et al., it is highly likely that the peninsula of Lista is strategically advantageous due its natural harbours and portages [34]. It would also lend itself as a natural resting place after crossing the Skagerrak strait and function as a staging ground for travels further up the coast or south to Jutland.

The strong connections between Thy and Lista in the Bronze Age is thus evidenced by the presence of imported lithics and metal artifacts from the Limfjord region, as well as notable similarities in burial practices in earthen mounds, cists made of stone slabs and similar architectural styles seen in three-aisled longhouses [4]. This similarity continuous west and

northwards along the Norwegian coast. In contrast, when facing eastwards from the Lista peninsula the material and architectonic similarities are less clear. Burials are here instead mainly built with stones instead of earth (so called stone cairns), which appear in the thousands along the paleogeographic archipelagos of southern Norway and western Sweden, the majority of which are believed to date to the early Bronze Age [41]. An otherwise denominative feature for all southern Scandinavia is the rock art which appears in clusters along these coastal waterways and which date to c. 1700 – 200 BC. The maritime location of this imagery is further emphasized by its predominant focus on boats [3,42,43].

Bohuslän, placed roughly halfway along the coastal route between Lista and Thy, is the richest rock art area in Europe and in Scandinavia, featuring over 10,000 boat images, and is believed to have been an important boatbuilding and transit area in the Bronze Age [44]. Relatively large concentrations of flint and sickle daggers as well as bronzes found in this region points towards significant and often intense communication with the Jutland also for this region from the late Neolithic and throughout the Bronze Age [20,31].

Thus, whereas rock art and general artefacts point towards the possibility that the two regions of Thy and Lista might have been communicating via the coastal route, the very specific similarities in certain types of materials and architectonic expressions might suggest the coastal route was bypassed altogether. So far, no studies have quantitatively modelled maritime travels using either route. Frode Kvalø [24] has done a thorough study on the possibilities of the direct routes across the Kattegat Strait based on oceanographic, ethnographic and experimental data; whereas, Bengtsson [20,43] instead has focused on potential seafaring routes between Bohuslän and Jutland, identifying the relative safety afforded to small open boats operating within archipelagos or along lee coast in relation to the direction and strength of the wind. However, there are now several new methodological and computational considerations that need to be evaluated, and which might help shed new light on the nature of maritime communication between the two regions in the Bronze Age.

### The sea routes – navigational aspects and considerations

In addition to Kvalø [24], direct routes between northern Denmark to southern Norway have been proposed by Marstrander [45], Johansen [46] and Østmo [19] with the three suggesting Hanstholm on the north-western point of Thy as the most likely departure point (Fig 2). The Hanstholm Knude, a 9 x 2 kilometres large chalk ridge that rises to a height of over 60 meters above the present-day sea level, provides a striking point of reference within the landscape, and an equally important point of reference for seafarers (Fig 3) [23].

Marine sediments indicate the hinterland of this chalk ridge might have been an archipelago in the Early Bronze Age and barrows located at the top and base of the Hanstholm Knude as well as a contemporary settlement at Bjerre Enge to the southeast of the ridge suggest the area was important at this time, perhaps because of the shelter afforded by these islands for seafarers intent on either crossing the Skagerrak to the north, for fishing activities, or as a first point of shelter for southbound sea crossings from the Lista region (Fig 3) [23,24,49,50]. For this reason, a stray find of a flanged axe (Fig 3.) dated to c. 1950–1700 BC in the northern part of Bjerre Enge is of interest, since it indicates that already at this early period the area was visited, when potentially the axe was dropped from a boat [23].

Whether this Littorina Sea in any way connected the Hanstholm area with the Limfjord area to the south remains uncertain, not least since erosion rates in the area are high (Fig 4). This uncertainty makes it imperative to consider the possibility that a seafarer intent on reaching Hanstholm from the Limfjord might have had to follow the unprotected west coast of Thy for some 50 kilometres (Fig 2). Present day erosion rates along this coast indicate that up to 5 kilometres of land has been lost since the Bronze Age [23]. This makes it difficult to

**A.** View to the north from the hinterland of Hanstholm which reaches an elevation of c. 30m in comparison to the former Littorina sea. Drawing R. Christensen 1886 (after Bech 2018).

**B.** Bulbjerg 1899; a 47m high rock providing a striking landmark. Birds nesting on the cliff might have aided seafarers to find land when visibility was poor (birdlandingplaces.eu).

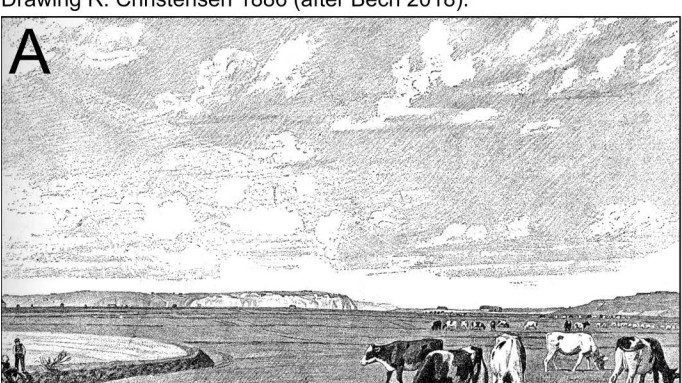

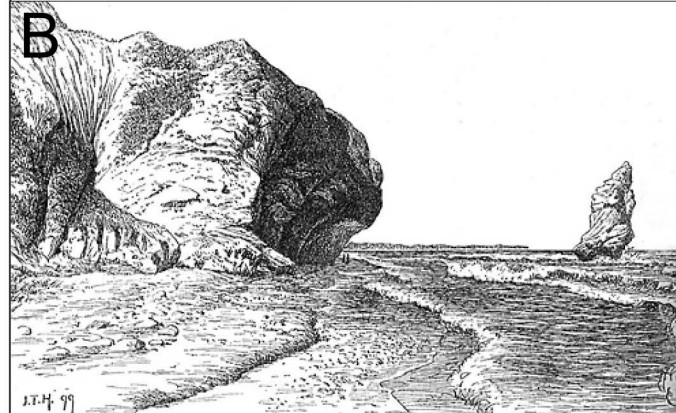

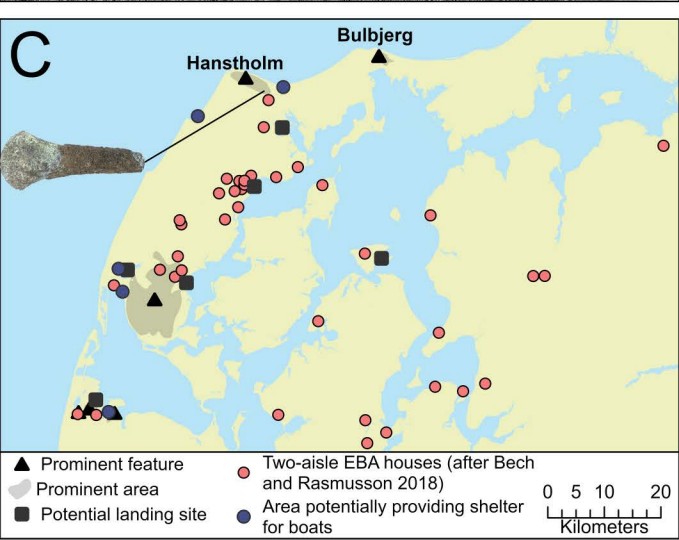

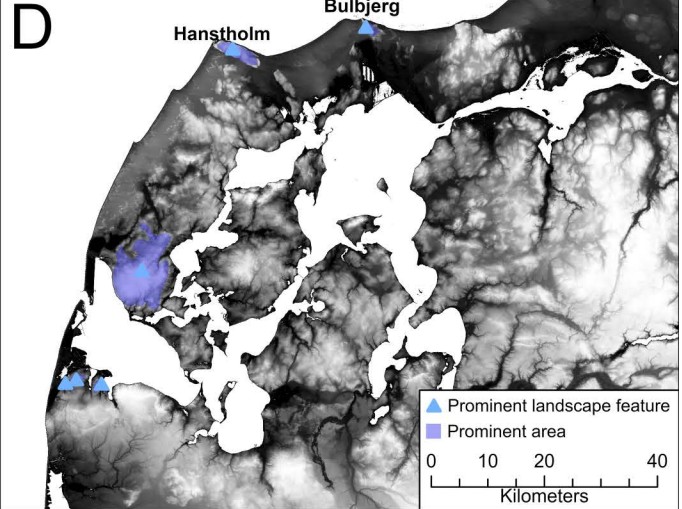

**C.** Flanged axe found during dredging in the northern part of Bjerre Enge. Votive deposit or dropped from a boat? Dated to c. 1950-1700 BC (Bech 2018, Vandkilde 1996). Map drawn by authors.

**D.** Current DEM highlighting the approximate location of sites in Figs 3A and 3B.

Maps contain DEMs and data derived from interpolated data from Sveriges geologiska undersökning (after Påsse and Daniels 2015) and Klimadatastyrelsen.

**Fig 3. Evidence of Early BA houses in relation to the contemporary shoreline, landmarks and other potential navigational aids in the Thy region** [23,47,48]**.** Maps created by Green and Bengtsson. Drawing of Hanstholm by R. Christiansen 1886 (reprinted with permission from Jysk Arkæologisk Selskab under CC BY 4.0 license). Drawing of Bulbjerg and Skarreklit by J. T Hansen 1899 (1848—1912), no copyright restrictions exist. Bronze Axe, (redrawn after Bech 2018). This map shows a paleoDEM created by the authors in ArcGIS Pro using elevation data from Klimadatastyrelsen under a CC-BY license and shore displacement values provided by SGU (after Påsse and Daniels 2015).

assess the availability of natural landing places along this route, perhaps afforded by small river inlets, or whether boats would have had to rely on being able to make the trip in one go or otherwise be dragged up on the beach well in advance of any bad weather fronts. Because of these uncertainties, we will focus on a slightly longer direct crossing straight from the mouth of the eastern entry to the Limfjord, as it is within the sheltered waters of the Limfjord that we find the main thrust of human occupation for this period as well as a multitude of potential landing sites near Early Bronze Age houses (Fig 3).

**Navigating the shorter direct route.** As the bird flies, the direct route between the mouth of the Limfjord across to Lista is around 180 km long (Fig 2). If first following the coast up to

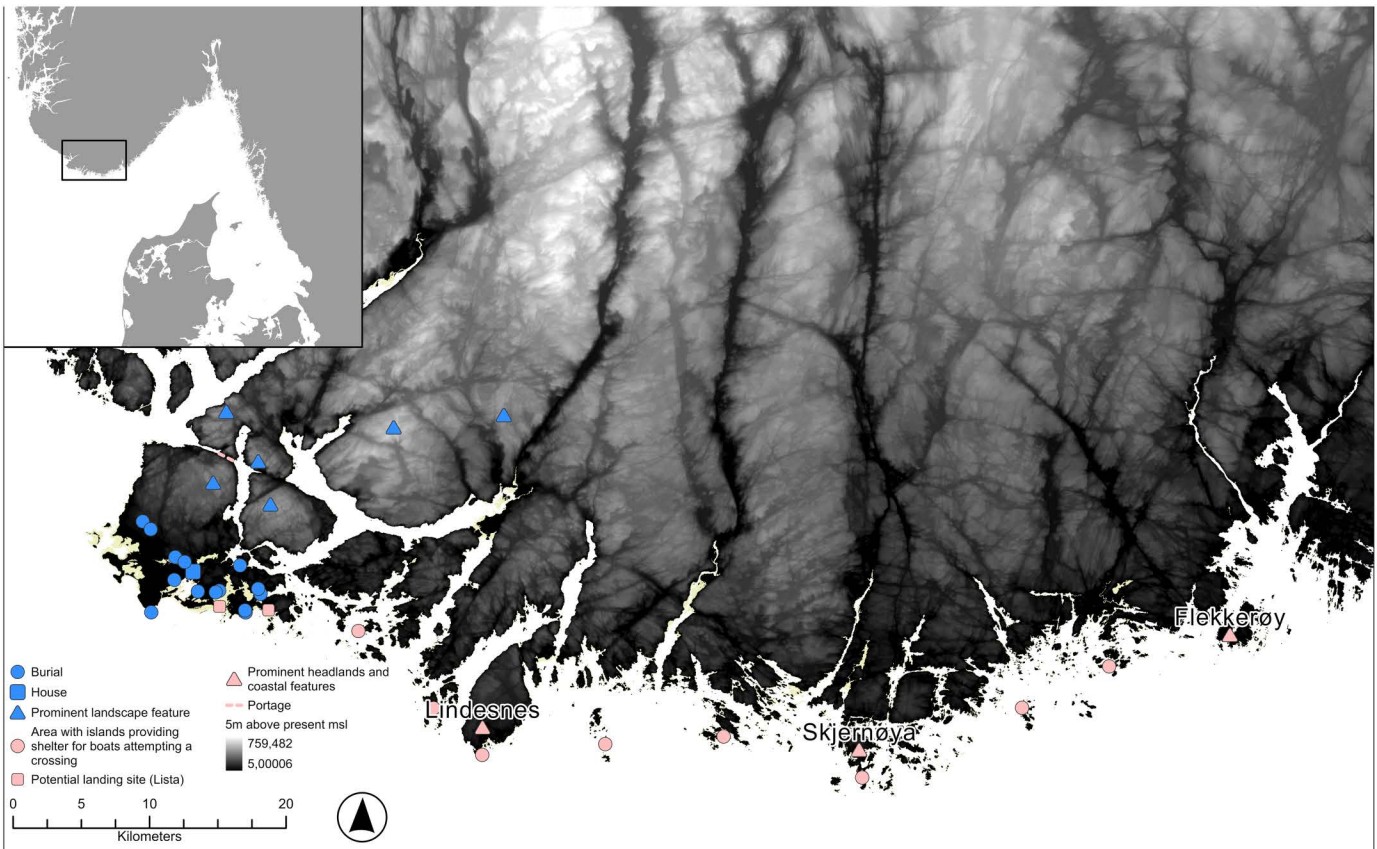

Maps contain DEMs and data derived from interpolated data from Sveriges geologiska undersökning (after Påsse and Daniels 2015) and Kartverket. Burials and houses after Austvoll (2020). Overview map uses GADM boundary data.

**Fig 4. A 1500 BC paleogeographic map of Lista overlaid by a map of evidence of early BA occupation in relation to the present day 5 meters above sea level (m.a.s.l.**) elevation curve (after [4]). The paleogeographic map does not consider wetlands and potential peat growth, nor does it show lakes and inland river systems. The topographic map provides an idea of the nature of the coast in terms of elevation and more striking points of reference for seafarers. This map shows a paleoDEM created by the authors in ArcGIS Pro using elevation data from Kartverket under a CC-BY license and shore displacement values provided by SGU (after Påsse and Daniels 2015).

Hanstholm, the direct crossing over open sea would be reduced to about 110 km in a north-north-westerly direction before landfall amongst the outer islands off the coast somewhere between Flekkerøy to the east and Skjernøya. However, if the weather was clear enough and the crew could manage it, it might aim straight for the Lindesnes promontory, which with its height of over 100 meters to this day afford a distinct coastal landmark to seafarers whilst at the same time offering a multitude of inlets that could provide shelter for a boat with limited draught [51] (Fig 4).

There are several scenarios when, primarily during the summer months when cloudiness and instances of mist are at a minimum [52], a direct crossing might have been feasible from either the Limfjord or Hanstholm, even in a relatively small boat using only very basic methods for navigation [53, 54]. Any crossing party would firstly lie in wait of favourable weather conditions, most likely guided by the reassurance of one or more consecutive red evening skies [51]. The red evening sky phenomenon is caused by the trapping of dust and particles in the atmosphere by high pressure [55]. This causes blue light to scatter, leaving the red light to colour the sky [55]. In regions such as Scandinavia, where weather systems tend to come from the west, this is a reliable indicator of the arrival of a high-pressure front bringing with

it fair weather for that night and the following day, perhaps even longer depending on how big the system is and how quickly it is moving. Thus, waiting for this phenomenon would be a good way to establish a relatively safe time of departure and would usually ensure a reliable morning breeze to help push the boat for the first 10 kilometres or so before petering out, but seafarers could equally wait for southerly winds to establish and depart hoping these will stay put for the duration of the trip. In whichever case, the crossing could hypothetically be initiated at any time from sunset.

In southern Scandinavia, twilight remain throughout the night in the months between late May until late July, whence the centre of the sun stays within 12 degrees below the horizon after sunset. During these months the sun sets in the NW and rises in the NE and the direction of the suns passage remain noticeable as a faint light across the horizon. After sunset there is almost an hour of good light (civil light), followed by a 3–4-hour long period of nautical light during which time the horizon is visible as well as the strongest stars [56, 57]. This should make it possible to navigate northwards on the sun on a clear night, in open sea with no obstacles, but would not be sufficient for safely navigating within an archipelago.

During darker nights, stars could potentially have provided basic guidance of direction at sea. The celestial north pole in 1500 BC was not marked by the current polar star, the Polaris, but the relative positions of stars around this point could have been used to determine a general direction of the north (compare with [58–60]). If not navigating at night, any crossing initiated from an hour before sunrise would be limited by the hours of light which range between 19 or 20 hours in the summer months to only 15 to 16 hours in April or October and barely eight during the darkest days of the year close to the winter solstice in late December.

Once the sun rises above the horizon it continues its use as a point of reference. Heading in a northerly direction it would be comparatively easy to keep track of its semicircular path across the sky, in comparison with the horizon or the rail of the boat and the wind or wave direction, in the knowledge that at its zenith, it appears to stand still for four minutes before continuing in a semicircular descendent [61]. Thus, during the months of the year with abundant sunlight, navigation would have been conducted primarily by the position of the sun. Other important points of reference would be the temperature, smell, salinity and colour of the sea and the direction and nature of the swells and/or waves when travelling across the Skagerrak, and even the sightings and flight paths of birds, perhaps in relation to known nesting cliffs [58]. The variation of the depth across the route would most probably have effect on the type of waves/swell and the colour of the sea, in addition to which currents carrying water that is warmer or less salty might be another point of reference, all of which a seasoned seafarer would have been observant of [58]. Between Jutland and southern Norway there are several very distinct "bands" of depth along the route, ranging from relatively shallow water on the first half, but including one section of water depth below 100 m, whereas the second half is characterised by depths of more than 200 m, before reaching the last band around 15 kilometres before landfall when water depths decrease to around 20-50 m (Fig 2).

Another important navigational aid are clouds [51]. The most important of these are the cauliflower shaped formations of cumulus clouds that usually form over land on fair-weather days once the sea breeze fills in from the early afternoon [20,51]. On a clear day such clouds can be seen over the relatively low-lying Jutland from over 70 kilometres (Figs 5 and 9). However, on particularly hot summers days, clouds might not form, in addition to which the sight conditions are generally less clear, impairing the sighting of land from the sea even further.

On a clear day, on the basis that the southern Norwegian coast features mountains with peaks in the region of 300 to over 600 meters height, this coast should be within sighting distance from the sea at around 35-45 kilometres before landfall even without a cloud cover [62]. That should allow a navigator to adjust the course towards the western part of

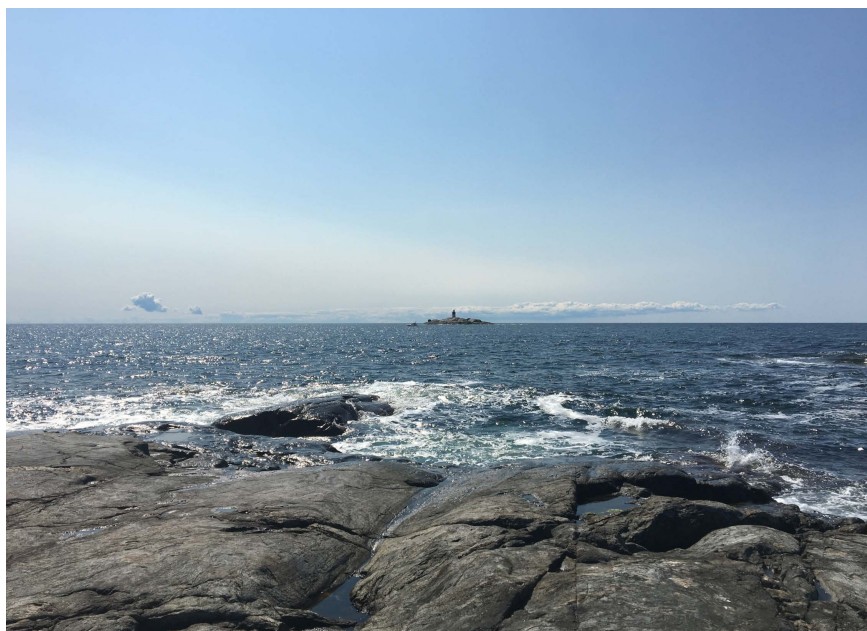

**Fig 5. Cumulus clouds forming over northern Denmark, as viewed from Lilla Pölsan (4 km due west of the islands of Rörö/Hönö) in the northern Gothenburg archipelago with the light house on Stora Pölsan in the foreground.** The distance from here to Denmark is c. 70 geoidal km and the wind speed at the time 4-5 m/s (photo: Boel Bengtsson, July 2020 at a height of c. 4 m.a.s.l.).

whatever land is visible. Once a particular landmark such as the Lindisnæs promontory has been identified, potentially once the vessel is within around 24 kilometres, further adjustments to the course can be made (Fig 6). In the opposite direction, land is much harder to detect from a distance unless cloud formations have developed, depending on which Hanstholm (or Bulbjerg) might not be detected until a vessel has come within a radius of c. 20 km (Fig 6).

Thus, a small vessel setting off from Jutland might expect to be out of sight of land after c. 20 km (but could potentially detect land behind if/once clouds have developed towards the afternoon), whereafter it would have to proceed using a combination of navigational aids provided by primarily the positioning of the sun and the direction of the swell, in combination with a sound knowledge of the boat and how it is affected by wind and waves (Fig 7) [53]. In clear weather the total length of open water navigation without sight of land in any direction is approximately 50 kilometres if no clouds have formed over land. On clear weather days where clouds do form over the high mountains of southern Norway, this distance might be greatly reduced (Fig 8) [55]. While allowing for reduced visibility, a lack of thermal cloud formations, and variable wind and sea conditions, navigational error (Fig 9), or expected radius of arrival in relation to the intended destination, might be in the region of 25 degrees. Ultimately, the success of this long open water journey would be down to a combination of skills including that of weather prediction and navigation, seamanship and the type and speed of vessel used [43].

**Navigating the longer coastal route.** The longer land hugging, or coastal "highway" route [20] suggested in this paper, which is a combination of mainly inland and coastal waterways with only two or three relatively short direct open-water crossings of up to 45 km length, is almost 600 kilometres longer than a direct route (Fig 10). From an environmental perspective, not considering any potential dangers or "costs" caused by social interactions, this route

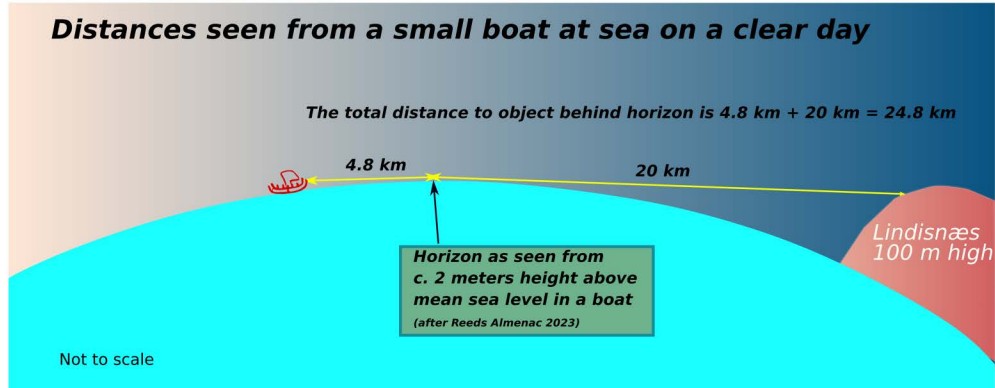

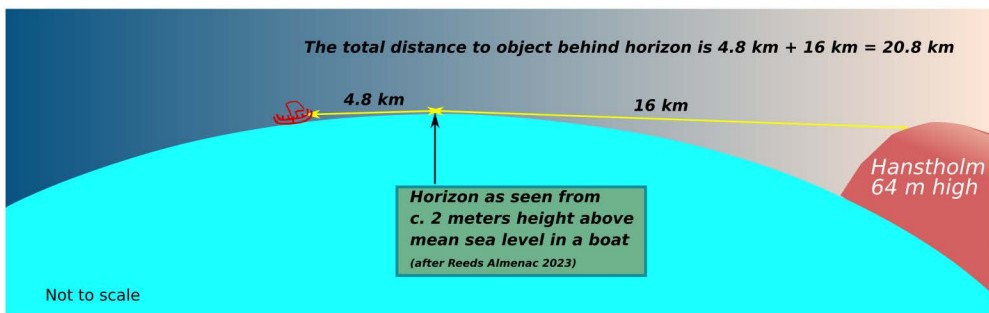

**Fig 6. Range of visibility of specific landmarks such as Lindinsnæs and Hanstholm** [62]**.** Visibility range based on a person standing in an open boat viewing the horizon from a height of approximately 2 meters above the sea.

has three main advantages in comparison to a direct Skagerrak crossing. Firstly, both the inland waterways of the Limfjord, the lee-coast of eastern Jutland and the archipelago that proceeds northwards from around the height of Onsala on the Swedish west coast from where it continues more or less all the way to Lista, provide a buffering effect on the amplitude of waves that can build up (see shaded areas along the sea route in Figs 1 and 10), the perhaps most important factor to consider when operating a small paddled (or oared) vessel (in addition to wind strength).

Wave height is commonly used to categorize commercial shipping zones for inland and coastal waterways (Table 1) and are based on the conditions of the main thoroughfares suitable for larger vessels but can be adopted to describe the significance the buffering effect has on small craft seafaring (Fig 11).

To a certain degree this buffering effect also applies to the average wind speed [64, 65] and would suggest windows of operation within a buffer zone, in particular for a vessel with a low draught, are much wider in comparison to areas outside this buffer zone where the maximum significant wave height is much higher (Fig 11).

Secondly, if excluding the shorter direct crossings, both the inland route through the Limfjord and the archipelagos offer not only a multitude of resting places along the route for a crew to recoup or await favourable conditions, but also a wider choice of routes to pick depending on the prevailing sea state and weather conditions. Thus, routes deep within the archipelago, nearest to the coast proper might have been chosen on windy days, avoiding any stretches of more open water, whereas portages might have been used to avoid particularly difficult stretches of coast (e.g., Lista). Portages could also have been used as short cuts across large headlands as and when needed (e.g., between Brastad and Kville – see B in

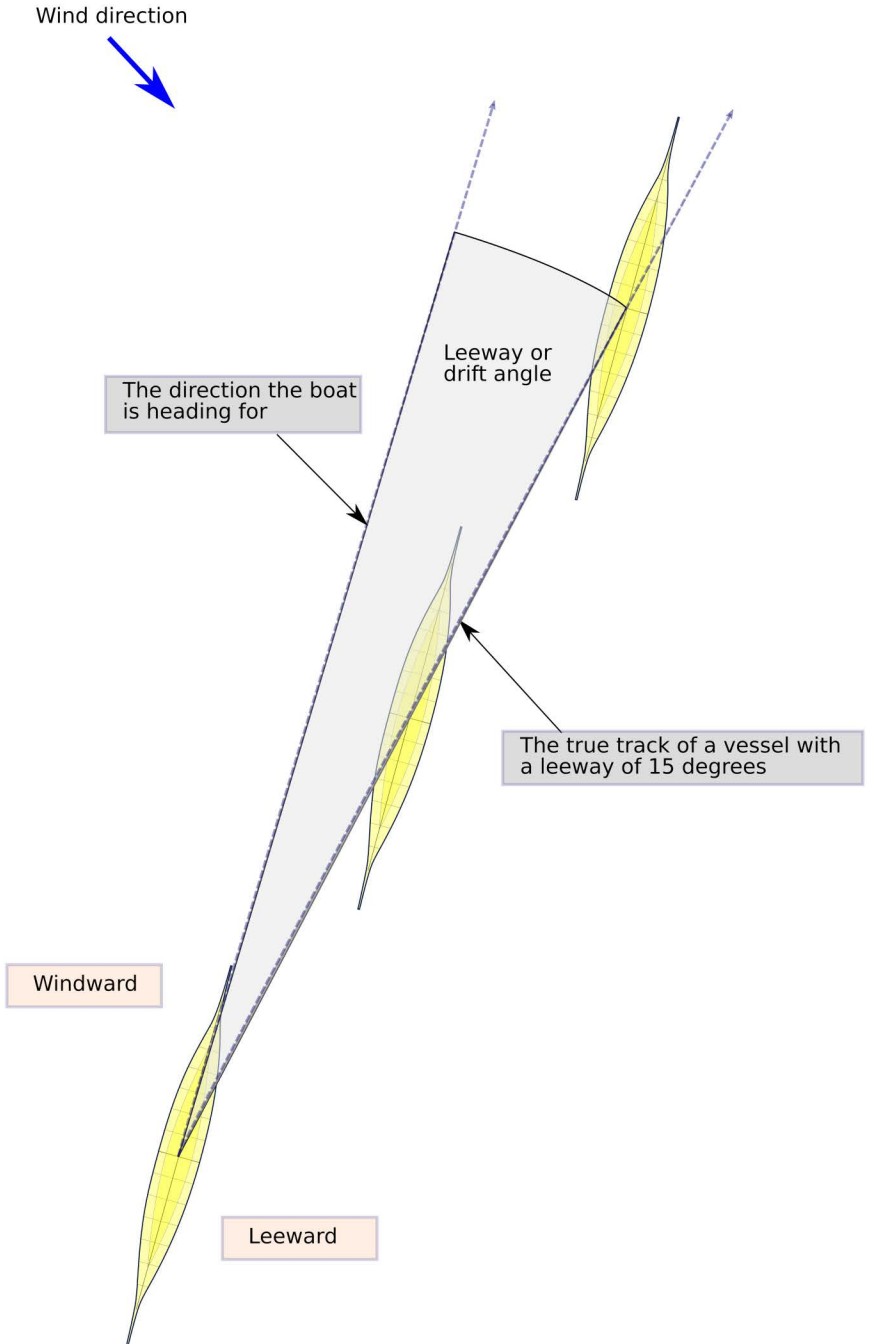

**Fig 7. The basic effect of drift or leeway due to wind on the course taken by a vessel.** This effect can be countered by, e.g., a keel or steering oars. Generally, the slower the boat the larger the drift. In a paddled boat, leeway will have to be countered by increasing the paddling effort on the leeward side or by reducing paddling effort on the windward side.

Fig 10). Ancient place names indicate locations where portaging was common practice and several rock art sites dating to the Bronze Age are located near such potential portage routes, a connection further emphasised by imagery appearing to depict boats that were carried or even pulled by animals across land [3,42,66]. Thirdly, a coast hugging route makes navigation easier since land is always within sight.

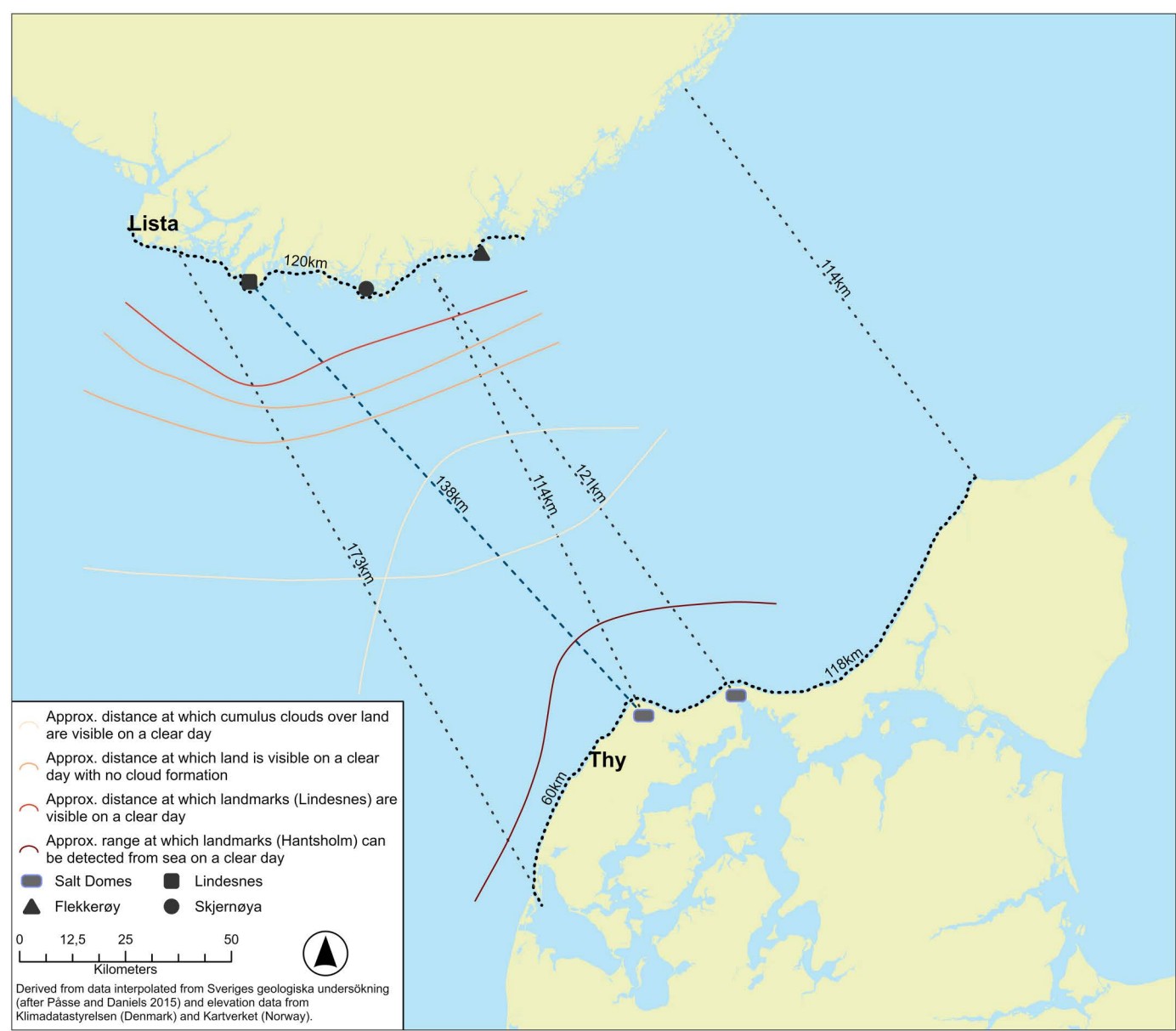

**Fig 8. Approximate ranges of land visibility for small boat approaching from the sea between Thy and Lista; landmarks, land in absence of cloud formations (two lines depending on level of visibility), and land with early thermal cloud formations forming above.** Figure by Ashely Green and Boel Bengtsson. This map shows a paleoDEM created by the authors in ArcGIS Pro using elevation data from Kartverket under a CC-BY license and shore displacement values provided by SGU (after Påsse and Daniels 2015). Note: None of the original elevation data is displayed.

Although the archipelago appears deceptively difficult to navigate for an outsider, knowledge about land formations, the ability to distinguish a headland from islands or which islands provide fresh water or safe havens, and the nature and location of hazards would most certainly be handed down and shared through the generations, in a similar way in which we today might provide instructions to a car driver (who does not have a GPS, map or a phone) (compare with [66, 67]). Whereas the many Bronze Age stone cairns that scatter the islands in the archipelagos along the route might have provided points of reference when navigating these waters depending on whether they were originally visible from the sea; they certainly

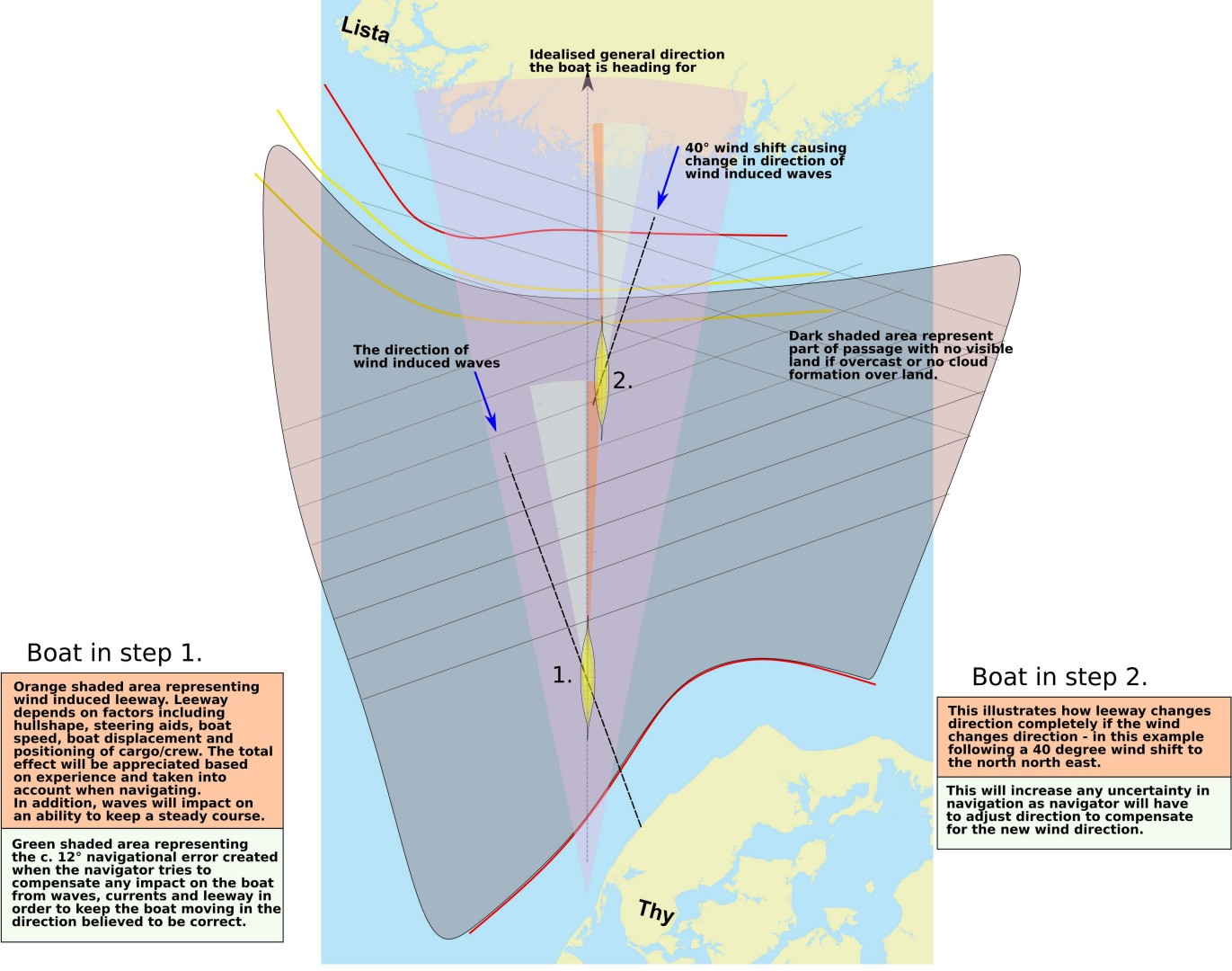

**Fig 9. Simplified illustration showing factors contributing to navigational error incurred by a navigator trying to reach a destination out of sight of land and with only basic navigational aids at disposal.** This map shows a paleoDEM created by the authors in ArcGIS Pro using elevation data from Klimadata-styrelsen and Kartverket under a CC-BY license and shore displacement values provided by SGU (after Påsse and Daniels 2015). Note: None of the original elevation data is displayed.

provide an appreciation of which regular routes seafarers were using [43,66]. Using such knowledge in combination with what is the most basic repertoire for any navigator – the use of thumbs, hands and arms to judge distance and course, and transit lines for entering or leaving particularly tricky areas - would have been enough to safely navigate these waters in the type of shallow drafted boats we believe were used in the Bronze Age [17,53,68].

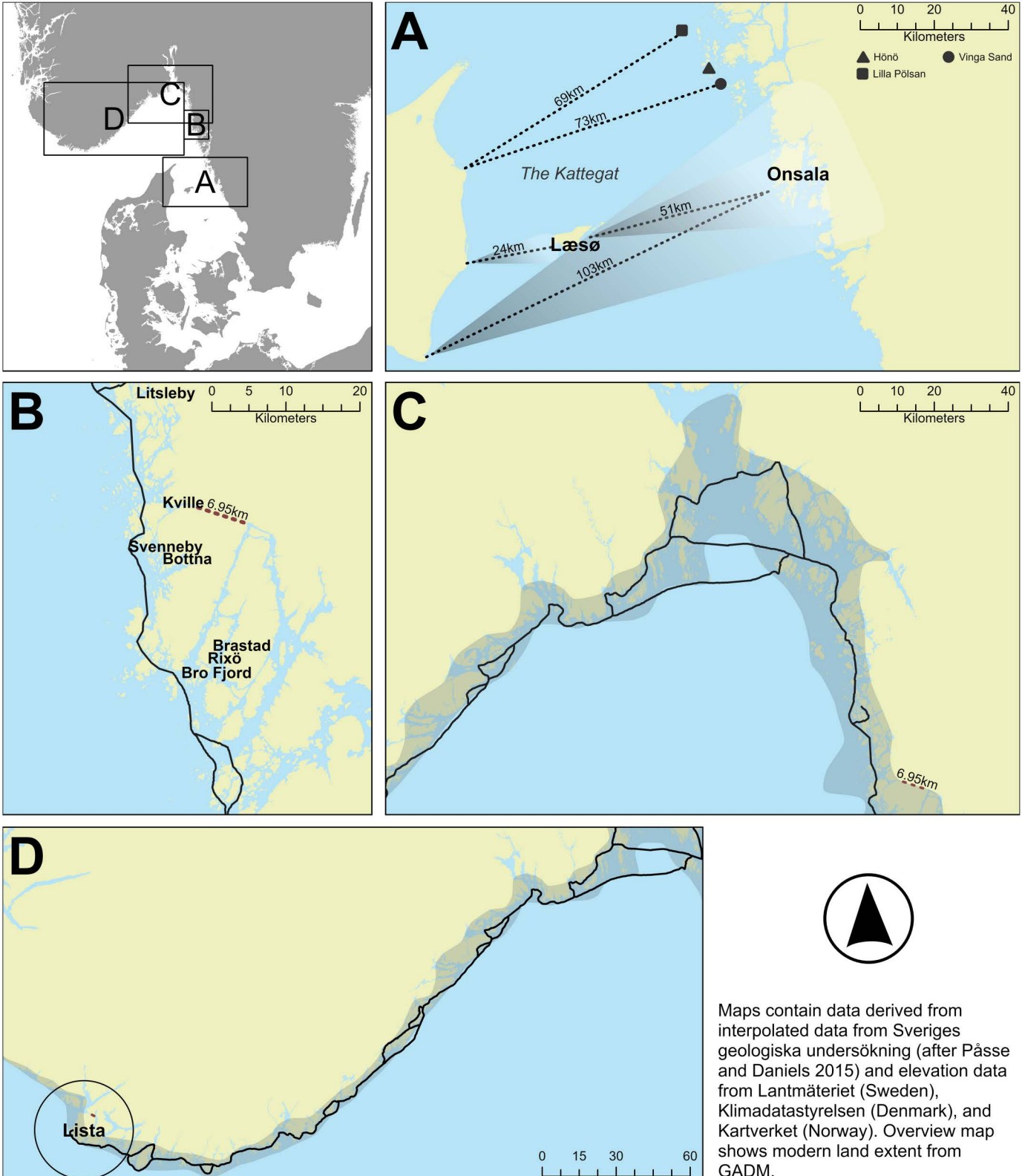

**Fig 10. A c. 1500 BC paleogeographic map of the longer route from Thy, passing through the Limfjord, the two or three more open sea crossings (shown in A and C in** Fig. 10**) and various potential routes through the archipelago on the coastal route towards Lista.** The paleogeographic map does not allow

for erosion rates, hence the exact configuration of Læsø and the water depths surrounding it are only an estimation. This map shows a paleoDEM created by the authors in ArcGIS Pro using elevation data from Lantmäteriet, Klimadatastyrelsen, and Kartverket under a CC-BY license and shore displacement values provided by SGU (after Påsse and Daniels 2015). Note: None of the original elevation data is displayed.

**Table 1. Examples of the categorization of inland and coastal waterways in Sweden based on the significant wave height, measured from base to crest of a wave, of the 10% largest waves observed within an area during a short period of time[63].**

|  | Significant Wave height | Examples of modern-day areas currently categorised |
|---|---|---|
| Zone 1 | never > 2 m. | Large inland lakes such as Lake Vänern and more open archipelagos (e.g., Brofjord and Donsö on the Swedish west coast) |
| Zone 2 | never >1.2 m. | Outer shipping routes within more a more open/narrow archipelago (e.g., btw Lysekil/Orust and Gothenburg southern archipelago) |
| Zone 3 | never > 0.6 m. | Wide rivers, inner archipelagos, water depth of 1.5 m or more |
| Zone 4 | No wave height | Narrow rivers with water depths below 1.5 meters. In-between islands/inner archipelagos |

As for the shorter direct routes within the longer coast hugging route, we suggest crossings might have been attempted taking advantage of the large shallow areas created by the Littorina Sea, at a juxtaposition of roughly 20 km from Jutland and perhaps 45 km from Onsala on the Swedish west coast, and from which the island of Læsø has formed (A in Fig 10). The oldest part of Læsø emerged as a c. 10 km long sand barrier spit around 2900 BC and appears to have remained similar in size until c. AD 1000 when it merged with a second raised sand barrier system[69]. High erosion rates make it difficult to confirm the exact configuration of the island in the intermediate periods but the presence of raised boulder reefs in the vicinity, and now submerged reefs marked out further to the north of Læsø on old maps [70], as well as the established merging of two separate spit islands to form the present shape of the island, would suggest spit islands might have appeared and disappeared throughout the intervening years without necessarily leaving any trace. This could explain why only archaeological remains related to Pitted Ware groups have been found on this oldest part of the island [71], whereas a lack of Bronze Age material might result from a change in the landscape and the loss of any landing sites used in the Bronze Age [66]. Whichever the case, these shallow waters would have ensured low wave amplitudes within the area, a perfect choice of route for shallow drafted vessels [20,43].

Thus, this longer land hugging route offers navigational challenges of a slightly different nature compared to the direct crossing of the Kattegat, in that it is not a "leap of faith" to same degree, and, although the same level of weather prediction/navigational skills and seamanship applies, the journey is not quite as dependant on the long window of "perfect" weather conditions in terms of waves and wind for other than much shorter periods. Therefore, on fair weather days, heralded by a red evening sky the previous evening, the crew can use the light winds in the morning for a more direct route while seeking more sheltered routes for the afternoon once the sea breeze has filled in, and which can often reach peaks of 8-10 m/s in the early afternoon [20].

## Producing a polar diagram of a Bronze Age (BA) type vessel

The simulations of potential routes suggested in this paper rely on available performance data of a Bronze Age type vessel. The only vessel that can be argued represent a Bronze Age type vessel [3,17] and for which such performance data exists that could be used is the c.350 BC Hjortspring boat. This boat was found during peat excavations in the Hjortspring bog on the island of Als in southern Denmark in the 1880's and was excavated in 1921–1922 [72]. About

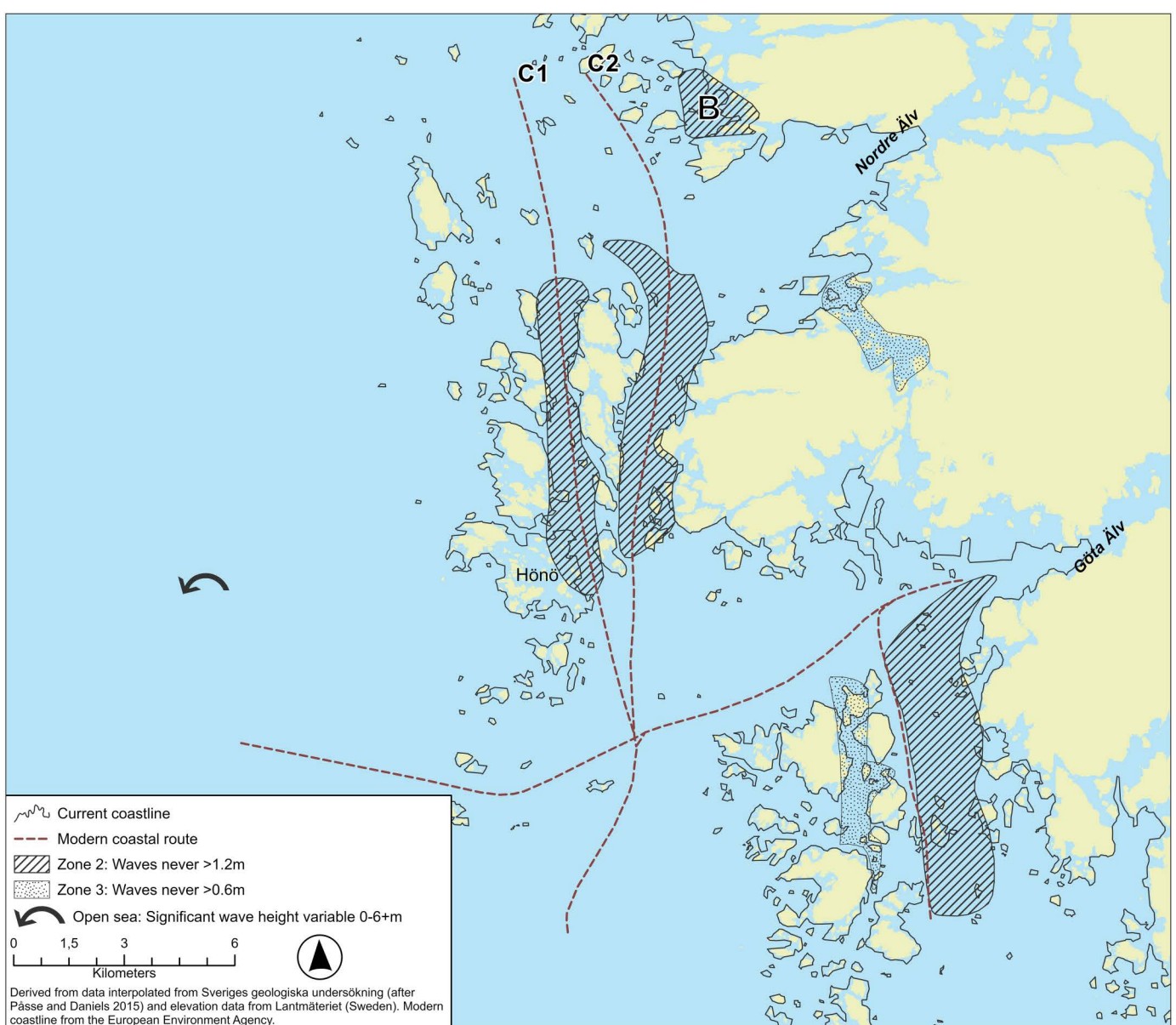

**Fig 11. A segment of the Gothenburg archipelago on the Swedish west coast showing an outline of the present coastline in comparison to that of c.** 1500 BC. Shaded zones provide examples of how segments within the paleo coast can be categorized in comparison to modern classification systems for commercial vessels operating in the area. In comparison with modern boats with deep draught, prehistoric vessels can safely make use of almost any segments of the environment. Zone marked B shows area where canoe trials were made in 2005, an area which today would be classified as a zone 3 but which was likely a zone 2/3 in 1500 BC [17]. This map shows a paleoDEM created by the authors in ArcGIS Pro using elevation data from Lantmäteriet under a CC-BY license and shore displacement values provided by SGU (after Påsse and Daniels 2015). Note: None of the original elevation data is displayed.

40% of the boat has been recovered, enabling the reconstruction of a double ended plank-built boat that from stem to stem is c. 14 meters long, with a total of 10 internal thwarts, each with carved out seats for two paddlers (Fig 12) [72, 75]. The overall length of the boat is extended by two sets of horn projections at either end. The lower of these are attached to the c. 15.4 m long bottom plank which protrudes from the bottom plank at each end, whereas the upper horn projections extend outward and upwards following the shape of the gunwale, making the

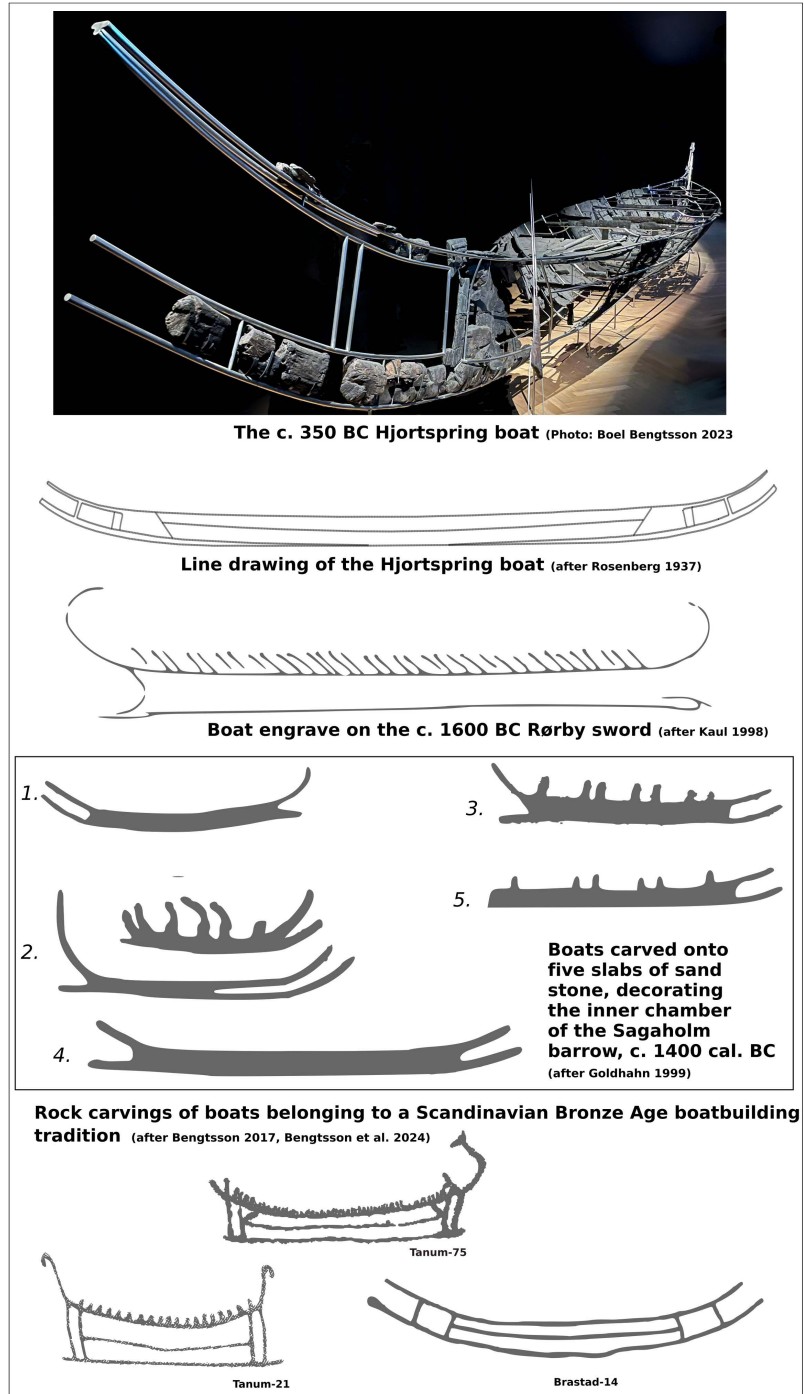

**Fig 12. A comparison between the Hjortspring boat** [72] **and southern Scandinavian boat depictions [redrawn after 3, 19, 73, 76].** Photo of the Hjortspring boat taken by Boel Bengtsson, Jan. 2023 and due for print under CC BY 4.0 license in Bengtsson 2025 [74].

total length of the boat around 18 m [76]. With the find, 16 individually made paddles were found with relatively narrow blades, making them ideal for long distance paddling [77], along with two steering oars, one located at each end of the vessel [72]. Neither of the two steering

oars were complete and estimates of their individual blade lengths vary between 53 cm [72] to 75 cm, but could have been longer still (Fig 13) [17].

Parallels to the unusual design features of the Hjortspring boat appear in depictions of boats in both rock art and bronzes dating from c. 1600 BC onwards in Scandinavia (Fig 12) and are also marked out on contemporary ship-settings in the region [3,20,42,78,79]. This, along with the refined boatbuilding technology employed in its construction, strongly suggests that it was built within a well-established Scandinavian boatbuilding tradition with its roots at the very beginning of the Bronze Age [3,20,53,80–82]. Hence it is justifiable to refer to it as a "Bronze Age Type Boat" despite it being of a slightly younger date [17,20].

A reconstruction of this boat, called the *Tilia Alsie*, was launched ready for sea trials in 1999 [76], and was, between 1999 and 2001, tested extensively by both members of the Hjortspringbådens Laug and professional Dragon boat racers under the supervision of Max Vinner from the Viking Ship Museum in Roskilde [83]. Both the process of reconstruction and the on-water trials and their results have been published in the Ships and Boats of the North series in a volume co-edited by Ole Crumlin-Pedersen and Athena Trakadas in 2003 [75]. Further testing of the vessel was made in 2006, this time under sail, the results of which were published in the Maritime Journal of Archaeology in 2011 [17].

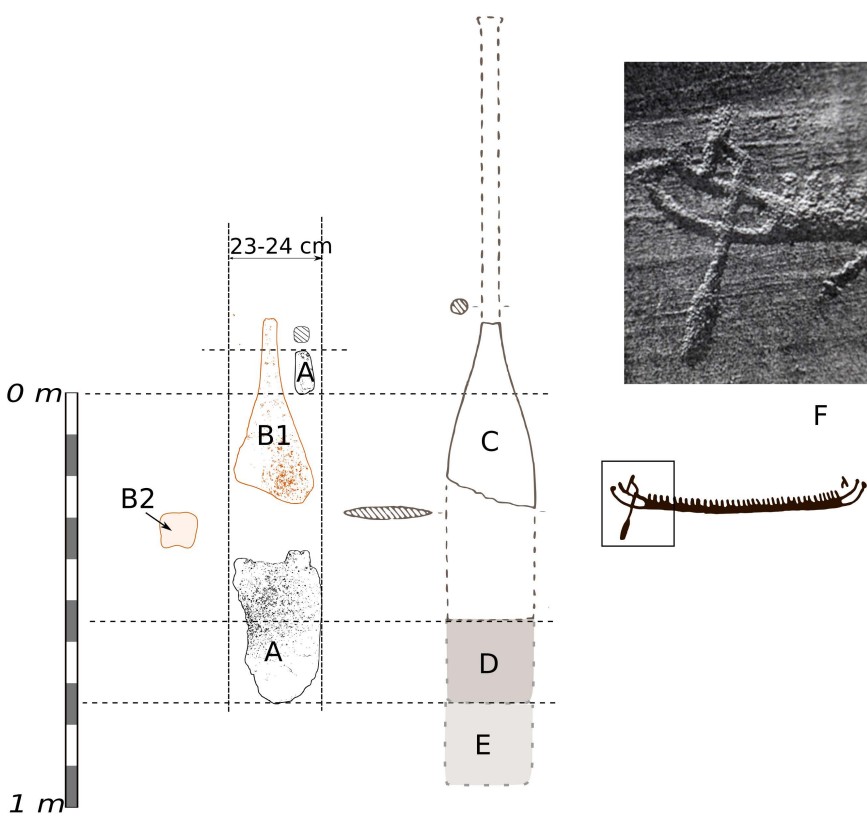

**Fig 13. A. remains of steering oar from the north end of the Hjortspring boat [72], B1 and B2, two pieces of a corresponding steering oar from the south end.** C, the steering oars as interpreted by Johannesen [72], C/D, size steering oars made for the Tilia Alsie [43]. E. Tilia Alsie's steering oar with a further 25 cm added depth. F. Detail of humanlike figure holding a steering oar, the Rixö rock art site in Bohuslän, Sweden (photo S.A Hallbäck, https://shfa.dh.gu.se/).

The performance data from the above publications has been collated and assessed for the purpose of making a series of polar diagrams, roughly equivalent to that of a BA type boat. This data on its own provides snippets of information of boat performance depending on the number of active paddlers and paddle strokes per minute, steering oar configuration and displacement. We are here using modern naval architect methods to fill in the gaps in this information to predict its performance across the full range of environmental conditions that can be expected when simulating long distance seafaring. This process has also depended on the availability of 3D-records of the *Tilia Alsie's* hull shape and further helped by the many detailed geometrical and hydrodynamic calculations recorded such as the *Tilia Alsie's* hull stability and resistance [75].

The *Tilia Alsie* varies from Johannessen's interpretation of the original Hjortspring vessel in that it has a more curved profile, resulting in a much shorter water line (Fig 14). The overall difference in performance between the two versions has not been calculated or compensated for in the polar chart presented here but suggests the Tilia Alsie is a directionally less stable vessel than the original Hjortspring boat [20,83].

## Velocity prediction processes applied to the BA type vessel

To predict the boat speeds of the vessel in a range of environmental (i.e., wind direction/speeds and waves) and loading condition (number of crew and quantities of cargo/stores), a software tool was developed. This was based on existing methods usually applied to modern sailing yachts and powered vessels, but with adaptation to accommodate the feature differences of the *Tilia Alsie* in comparison to modern yachts.

The paddling and sailing aspects were modelled with different models within the overall tool, though many of the components are common between the two approaches. This paper will focus on the paddling model.

**Paddling model.** The paddling model consists of the balance of force components in the boat track axis and perpendicular to in the waterplane (i.e., side force), essentially, balancing the paddler thrust contribution to the opposed resistance components to obtain a boat speed for each combination of true wind speed and direction. For this the following forces are included:

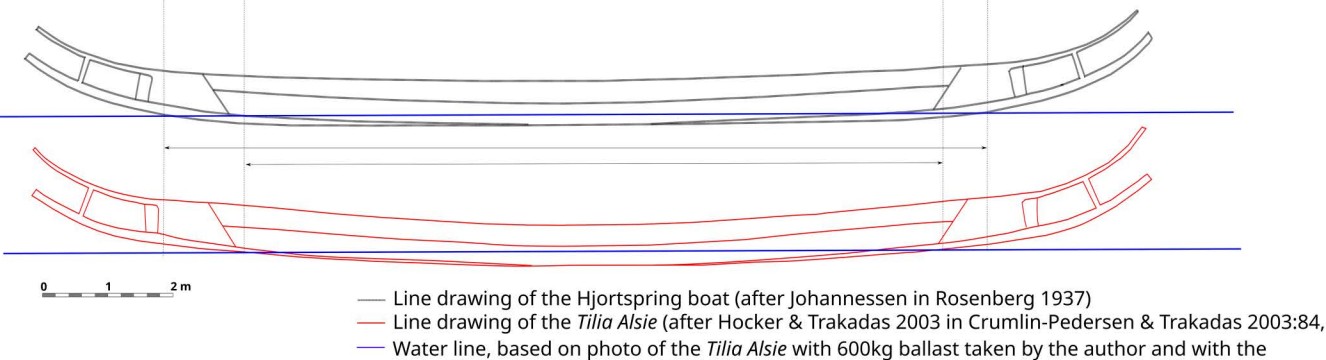

—— Line drawing of the Hjortspring boat (after Johannessen in Rosenberg 1937)
—— Line drawing of the *Tilia Alsie* (after Hocker & Trakadas 2003 in Crumlin-Pedersen & Trakadas 2003:84, 91)
—— Water line, based on photo of the *Tilia Alsie* with 600kg ballast taken by the author and with the water line for Johannessen's Hjortspring boat adjusted according to the principles of Archimedes

Difference in the hull shape between the original Hjortspring boat and the *Tilia Alsie* reconstruction:
*Tilia Alsie*: less course stability due to shorter water line. Needs cargo or double steering oars to compensate for this (create a longer water line).
Hjortspring has a longer water line and more of the V-shaped endships aiding course stability.
*Tilia Alsie*: lower side but easier to turn.

**Fig 14. The Tilia Alsie in comparison to the interpretation of the original Hjortspring boat [72,84].**

**1. Hydrodynamic.**

- Viscous (primarily frictional) drag of the hull (vessel body) and any steering oars. Here the viscous drag is estimated using procedures based on the ITTC-1957 friction correlation line, with the additional for appendages, namely the steering boards, with the addition of form effects using the Hoerner 1965 formulations [85].

- Residuary (wavemaking) drag of the vessel body. This is estimated using the procedures detailed in Jackson 1995 [86], based on data for kayak type hull forms. The dimensional parameters of this vessel are broadly within the regression limits of this approach.

- Additional drag due to surface roughness/finish. Here an allowance was made for the surface roughness of the stitched wood plank finish based on sources such as Hoerner 1965 [85].

- Added resistance in waves. The added resistance in waves (Raw) used the Delft regression [87]. This approach used inputs of displacement, waterline length, pitch inertia and significant wave height to predict the additional resistance of the vessel in a seaway. With the *Tilia Alsie* vessel parameters of displacement/length ratio fitting within the regression's range. The significant wave height input was estimated from a function based on true wind speed from the Douglas Sea State Scale. The Raw was modified to incorporate the head sea wave angles not included in the Delft report.

- Induced drag due to sideforce to resist windage transverse component. The induced drag resulted from the vessel attaining an angle of attack to the flow to create hydrodynamic sideforce to balance/counteract the aerodynamic sideforce needed to maintain a steady course. This was estimated using a method from Van Oossanen 1980 [88] for sailing yacht canoe bodies without appendages and a drag model based on foil aspect ratio for the steering oars, with the model also including the effects of stall at large angles of incidence.

- Paddle force generated by paddlers. Here the paddle propulsive model used data recorded as a basis and this has been scaled based on cadence and boat speed for an individual paddler contribution and multiplied by the appropriate number of paddlers to estimate the total effective thrust [89].

**2. Aerodynamic.**

- Windage of both the hull and the crew. The aerodynamic windage model estimated the drag and lift (wind axis) of hull and crew, for a range of apparent wind angles and speeds. The hull was modelled using a standard ship based windage model, with the crew modelled as interacting objects (and a function of crew number) which varies with apparent wind angle. These were converted into boat axis for combining in the iterative force balance scheme with the other force components.

A 3D hull model was used to verify principal parameters for input into regressions and assess hydrostatics and stability. This provided inputs for various force component calculations so that a range of user defined loading and setup conditions could be tested, as can be seen in Fig 15.

## Assumptions and limitations of the paddling polar diagram used in simulations

It is important to bear in mind that the polar diagrams presented here are prediction based on data sources in combination standard naval architectural procedures. Greater confidence in the results would be achieved using high-fidelity approaches such as Computational Fluids

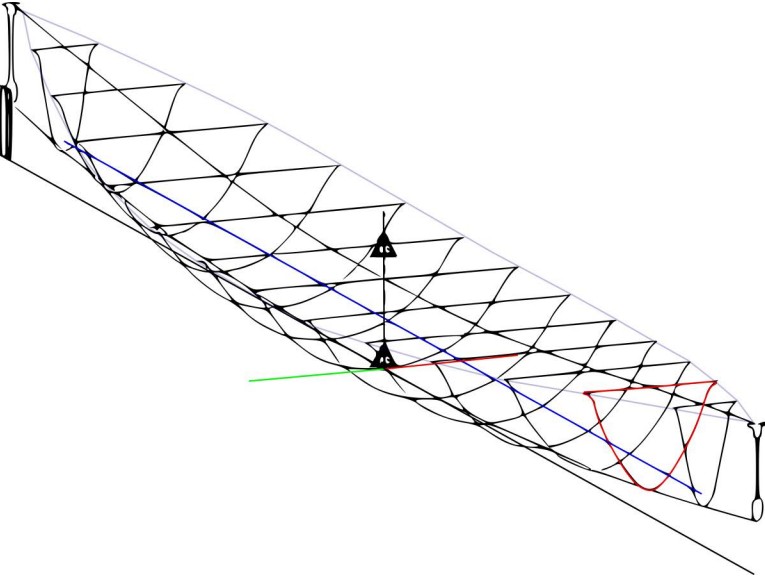

**Fig 15. Hydrostatic model of vessel.** Typical result outputs are provided in Fig 16 showing the predictions of paddling boat speeds with true wind speed and direction, in this case depending on the number of steering oars used.

Dynamics (CFD) and model scale tank testing as inputs to the model(s) or by making further on water tests in either the *Tilia Aslie* or a reconstruction of its hull shape [17,90].

Nevertheless, the predicted polar diagram used here is based on several assumptions.

(1) An active paddling crew of 16 with an additional four taking turns paddling. During sea trials under Max Vinner, this was deemed the most efficient crew strategy for long-distance paddling [83] (pers. Communication Knut Valbjørn and Niels Peter Fenger 2022/2023) and tallies the number of paddles found with the Hjortspring boat.

(2) Paddlers capable of paddling non-stop at a cadence of 50 for up to 20 hours when engaged in a direct sea crossing in ideal conditions. Again, these estimates are based on actual sea trials and what the competitive dragon boat paddlers engaged deemed they would have been capable of [83].

(3) The vessel is carrying cargo at its maximum displacement, which trial suggest lies around 3000 kg [83].

(4) The use of two steering oars to maximise the range of operational conditions (Figs 13, 16) [17,83].

Further to the above operational assumptions there are some immediate limitations that must be considered when using predicted paddled (or rowed) boat performance data for simulations. This primarily include the impact of leeway and waves on crew fatigue and the ability to keep an even pace over long distances under variable environmental conditions. There are unfortunately no adequate data for this since, e.g., the *Tilia Alsie* has mainly been used in very sheltered waters in maximum winds of c. 10 knots (pers. Comm. Valbjørn 2022).

Leeway occurs when a boat is travelling at an angle to the wind which pushes it sideways (Fig 7). Here the air resistance created by a paddling crew seated as high up as it is in the *Tilia Alsie* (Fig 17), is considerable, and naturally contributes to leeway in a calculative way, as such this is reflected in the polar diagrams in Fig 16. The predicted boat performance in Fig 16

**No steering oars**

Paddling Boat Speed (knots)

| True Wind Angle (degrees) | True Wind Speed (knots) | | | | | | |
|---|---|---|---|---|---|---|---|
| | 0 | 5 | 10 | 15 | 20 | 25 | 30 |
| 0 | 4.40 | 4.23 | 2.99 | 1.76 | 1.34 | 1.34 | 1.38 |
| 10 | 4.40 | 4.23 | 2.98 | 1.75 | 1.31 | 1.29 | 1.24 |
| 20 | 4.40 | 4.23 | 2.99 | 1.73 | 1.66 | 1.09 | — |
| 30 | 4.40 | 4.23 | 2.98 | 1.69 | 1.24 | — | — |
| 40 | 4.40 | 4.23 | 2.96 | 1.63 | 1.04 | — | — |
| 50 | 4.40 | 4.24 | 3.08 | 1.65 | — | — | — |
| 60 | 4.40 | 4.26 | 3.37 | 1.72 | — | — | — |
| 70 | 4.40 | 4.28 | 3.70 | 1.87 | — | — | — |
| 80 | 4.40 | 4.31 | 3.98 | 2.27 | — | — | — |
| 90 | 4.40 | 4.33 | 4.09 | 2.60 | — | — | — |
| 100 | 4.40 | 4.36 | 4.19 | 3.05 | — | — | — |
| 110 | 4.40 | 4.39 | 4.28 | 3.53 | — | — | — |
| 120 | 4.40 | 4.41 | 4.35 | 3.90 | 1.29 | — | — |
| 130 | 4.40 | 4.42 | 4.39 | 4.16 | 1.73 | — | — |
| 140 | 4.40 | 4.44 | 4.43 | 4.37 | 3.94 | — | — |
| 150 | 4.40 | 4.44 | 4.47 | 4.56 | 4.62 | 4.31 | — |
| 160 | 4.40 | 4.45 | 4.52 | 4.68 | 4.90 | 5.11 | 5.26 |
| 170 | 4.40 | 4.45 | 4.53 | 4.71 | 4.96 | 5.27 | 5.61 |
| 180 | 4.40 | 4.45 | 4.52 | 4.69 | 4.96 | 5.28 | 5.65 |

Leeway or Drift Angle (deg)

| True Wind Angle (degrees) | True Wind Speed (knots) | | | | | | |
|---|---|---|---|---|---|---|---|
| | 0 | 5 | 10 | 15 | 20 | 25 | 30 |
| 0 | 0 | 0 | 0 | 0 | 0 | 0 | 0 |
| 10 | 0 | 0.49 | 1.97 | 4.61 | 7.39 | 10.59 | 14.24 |
| 20 | 0 | 0.93 | 3.29 | 7.78 | 11.83 | 19 | - |
| 30 | 0 | 1.27 | 4.28 | 10.32 | 17.41 | - | - |
| 40 | 0 | 1.53 | 5.07 | 12.47 | 22.27 | - | - |
| 50 | 0 | 1.7 | 5.43 | 13.99 | - | - | - |
| 60 | 0 | 1.8 | 5.34 | 14.79 | - | - | - |
| 70 | 0 | 1.83 | 5 | 14.76 | - | - | - |
| 80 | 0 | 1.79 | 4.56 | 13.18 | - | - | - |
| 90 | 0 | 1.7 | 4.31 | 11.63 | - | - | - |
| 100 | 0 | 1.55 | 3.98 | 9.43 | - | - | - |
| 110 | 0 | 1.36 | 3.59 | 7.76 | - | - | - |
| 120 | 0 | 1.12 | 3.14 | 6.3 | 24.36 | - | - |
| 130 | 0 | 0.87 | 2.65 | 5.2 | 18.65 | - | - |
| 140 | 0 | 0.61 | 2.15 | 4.2 | 7.2 | - | - |
| 150 | 0 | 0.37 | 1.62 | 3.21 | 4.92 | 7.35 | - |
| 160 | 0 | 0.18 | 1.09 | 2.27 | 3.4 | 4.5 | 5.65 |
| 170 | 0 | 0.06 | 0.55 | 1.3 | 1.99 | 2.67 | 3.26 |
| 180 | 0 | 0 | 0 | 0 | 0 | 0 | 0 |

**One aft steering oar**

Paddling Boat Speed (knots)

| True Wind Angle (degrees) | True Wind Speed (knots) | | | | | | |
|---|---|---|---|---|---|---|---|
| | 0 | 5 | 10 | 15 | 20 | 25 | 30 |
| 0 | 4.38 | 4.20 | 2.98 | 1.76 | 1.45 | 1.34 | 1.38 |
| 10 | 4.38 | 4.20 | 2.98 | 1.76 | 1.44 | 1.31 | 1.29 |
| 20 | 4.38 | 4.21 | 3.00 | 1.76 | 1.40 | 1.14 | - |
| 30 | 4.38 | 4.21 | 3.02 | 1.76 | 1.30 | - | - |
| 40 | 4.38 | 4.21 | 3.02 | 1.73 | - | - | - |
| 50 | 4.38 | 4.23 | 3.17 | 1.80 | - | - | - |
| 60 | 4.38 | 4.25 | 3.51 | 2.06 | - | - | - |
| 70 | 4.38 | 4.27 | 3.84 | 2.68 | - | - | - |
| 80 | 4.38 | 4.29 | 4.11 | 3.71 | - | - | - |
| 90 | 4.38 | 4.32 | 4.21 | 3.96 | - | - | - |
| 100 | 4.38 | 4.34 | 4.28 | 4.10 | - | - | - |
| 110 | 4.38 | 4.36 | 4.34 | 4.22 | - | - | - |
| 120 | 4.38 | 4.38 | 4.38 | 4.31 | 1.75 | - | - |
| 130 | 4.38 | 4.40 | 4.40 | 4.38 | 4.28 | - | - |
| 140 | 4.38 | 4.41 | 4.43 | 4.48 | 4.54 | - | - |
| 150 | 4.38 | 4.42 | 4.46 | 4.60 | 4.81 | 4.96 | - |
| 160 | 4.38 | 4.42 | 4.49 | 4.68 | 4.95 | 5.27 | 5.61 |
| 170 | 4.38 | 4.42 | 4.50 | 4.68 | 4.95 | 5.28 | 5.66 |
| 180 | 4.38 | 4.42 | 4.49 | 4.67 | 4.93 | 5.25 | 5.63 |

Leeway or Drift Angle (deg)

| True Wind Angle (degrees) | True Wind Speed (knots) | | | | | | |
|---|---|---|---|---|---|---|---|
| | 0 | 5 | 10 | 15 | 20 | 25 | 30 |
| 0 | 0.00 | 0.00 | 0.00 | 0.00 | 0.00 | 0.00 | 0.00 |
| 10 | 0.00 | 0.17 | 0.83 | 2.46 | 5.30 | 8.55 | 12.41 |
| 20 | 0.00 | 0.32 | 1.40 | 4.57 | 9.20 | 17.48 | - |
| 30 | 0.00 | 0.46 | 1.76 | 7.26 | 12.02 | - | - |
| 40 | 0.00 | 0.57 | 2.04 | 9.64 | - | - | - |
| 50 | 0.00 | 0.65 | 2.19 | 10.95 | - | - | - |
| 60 | 0.00 | 0.69 | 2.11 | 10.56 | - | - | - |
| 70 | 0.00 | 0.71 | 1.92 | 7.51 | - | - | - |
| 80 | 0.00 | 0.69 | 1.76 | 3.94 | - | - | - |
| 90 | 0.00 | 0.65 | 1.68 | 3.21 | - | - | - |
| 100 | 0.00 | 0.58 | 1.57 | 2.88 | - | - | - |
| 110 | 0.00 | 0.49 | 1.43 | 2.57 | - | - | - |
| 120 | 0.00 | 0.40 | 1.26 | 2.26 | 18.98 | - | - |
| 130 | 0.00 | 0.30 | 1.07 | 1.94 | 3.19 | - | - |
| 140 | 0.00 | 0.21 | 0.85 | 1.63 | 2.48 | - | - |
| 150 | 0.00 | 0.12 | 0.61 | 1.28 | 1.84 | 2.42 | - |
| 160 | 0.00 | 0.06 | 0.39 | 0.90 | 1.35 | 1.70 | 2.19 |
| 170 | 0.00 | 0.02 | 0.19 | 0.47 | 0.77 | 1.06 | 1.29 |
| 180 | 0.00 | 0.00 | 0.00 | 0.00 | 0.00 | 0.00 | 0.00 |

**Two steering oars, one at each end**

Paddling Boat Speed (knots)

| True Wind Angle (degrees) | True Wind Speed (knots) | | | | | | |
|---|---|---|---|---|---|---|---|
| | 0 | 5 | 10 | 15 | 20 | 25 | 30 |
| 0 | 4.32 | 4.15 | 2.96 | 1.75 | 1.34 | 1.34 | 1.38 |
| 10 | 4.32 | 4.15 | 2.96 | 1.75 | 1.32 | 1.31 | 1.31 |
| 20 | 4.32 | 4.15 | 2.98 | 1.76 | 1.71 | 1.18 | - |
| 30 | 4.32 | 4.15 | 3.00 | 1.76 | 1.91 | - | - |
| 40 | 4.32 | 4.16 | 3.01 | 1.76 | 1.19 | - | - |
| 50 | 4.32 | 4.17 | 3.15 | 1.88 | - | - | - |
| 60 | 4.32 | 4.19 | 3.47 | 2.21 | - | - | - |
| 70 | 4.32 | 4.21 | 3.80 | 2.77 | - | - | - |
| 80 | 4.32 | 4.23 | 4.06 | 3.65 | - | - | - |
| 90 | 4.32 | 4.26 | 4.15 | 3.90 | - | - | - |
| 100 | 4.32 | 4.28 | 4.22 | 4.05 | - | - | - |
| 110 | 4.32 | 4.31 | 4.28 | 4.17 | 1.90 | - | - |
| 120 | 4.32 | 4.33 | 4.32 | 4.25 | 4.09 | - | - |
| 130 | 4.32 | 4.34 | 4.34 | 4.33 | 4.23 | - | - |
| 140 | 4.32 | 4.35 | 4.36 | 4.42 | 4.50 | 4.52 | - |
| 150 | 4.32 | 4.36 | 4.40 | 4.54 | 4.75 | 5.00 | 5.24 |
| 160 | 4.32 | 4.36 | 4.43 | 4.62 | 4.89 | 5.21 | 5.58 |
| 170 | 4.32 | 4.36 | 4.44 | 4.62 | 4.89 | 5.22 | 5.60 |
| 180 | 4.32 | 4.36 | 4.43 | 4.61 | 4.87 | 5.19 | 5.56 |

Leeway or Drift Angle (deg)

| True Wind Angle (degrees) | True Wind Speed (knots) | | | | | | |
|---|---|---|---|---|---|---|---|
| | 0 | 5 | 10 | 15 | 20 | 25 | 30 |
| 0 | 0.00 | 0.00 | 0.00 | 0.00 | 0.00 | 0.00 | 0.00 |
| 10 | 0.00 | 0.10 | 0.54 | 1.86 | 3.97 | 6.89 | 10.30 |
| 20 | 0.00 | 0.20 | 0.97 | 3.29 | 6.19 | 15.79 | - |
| 30 | 0.00 | 0.28 | 1.28 | 4.79 | 9.02 | - | - |
| 40 | 0.00 | 0.35 | 1.48 | 6.32 | 18.79 | - | - |
| 50 | 0.00 | 0.40 | 1.56 | 7.40 | - | - | - |
| 60 | 0.00 | 0.43 | 1.50 | 6.78 | - | - | - |
| 70 | 0.00 | 0.44 | 1.39 | 4.34 | - | - | - |
| 80 | 0.00 | 0.43 | 1.28 | 2.44 | - | - | - |
| 90 | 0.00 | 0.40 | 1.21 | 2.13 | - | - | - |
| 100 | 0.00 | 0.36 | 1.12 | 1.93 | - | - | - |
| 110 | 0.00 | 0.30 | 1.00 | 1.77 | 16.65 | - | - |
| 120 | 0.00 | 0.24 | 0.86 | 1.60 | 2.51 | - | - |
| 130 | 0.00 | 0.18 | 0.70 | 1.41 | 2.13 | - | - |
| 140 | 0.00 | 0.13 | 0.54 | 1.16 | 1.71 | 2.37 | - |
| 150 | 0.00 | 0.08 | 0.39 | 0.88 | 1.33 | 1.67 | 2.02 |
| 160 | 0.00 | 0.04 | 0.24 | 0.58 | 0.92 | 1.23 | 1.46 |
| 170 | 0.00 | 0.01 | 0.12 | 0.29 | 0.49 | 0.69 | 0.87 |
| 180 | 0.00 | 0.00 | 0.00 | 0.00 | 0.00 | 0.00 | 0.00 |

Legend:
- Leeway above 1 degree
- Paddling starting to become difficult
- Paddling not possible
- Drift starting to be a concern
- Large drift, boat difficult or impossible to handle

**Fig 16. Paddle prediction output for a BA type vessel with a total displacement of 3000 kg, using 16 active paddlers, paddling at a cadence of 50 strokes per minute.** The three outputs illustrate difference in speed (above) and leeway drift (below) for the boat using no steering oar, using one aft steering oar and with two steering oars, one at either end, with interpretation. Knots convert into m/s by being multiplied by 0.51.

shows, for example, that leeway of 1 degree occurs already in winds below 5 knots when no steering oar is used. With double steering oars this threshold is more than doubled. Thus, even relatively small appendages such as the two steering oars found with the Hjortspring boat have a large effect on its boat performance, allowing more ease of operation through a considerably larger set of environmental conditions than no steering or only an aft steering oar. With supporting evidence of its use provided by boat iconography as well as the experience of its significant benefit during sea trials [83], we presume two steering oars would have been the norm for any long-distance seafaring venture. By studying the increase in leeway and decrease in speed in the polar chart (Fig 14) and combining this with hands-on experience from on-water trials it is possible to estimate reasonable limitations in a crew's ability to maintain course stability and speed.

Because of these limitations, we apply an upper survival limit for waves at 2 meters. A survival limit indicates when a boat can no longer be operated and crew effort is entirely focused on making the vessel stay afloat (for more information on this, see Fig 17). As for wind strength, it is assumed that open sea crossings would only be initiated in conditions when the wind is below c. 5 knots (c. 3 m/s) and that at around c. 12 knots when leeway become more noticeable and require more paddling effort, the crew start choosing a course that is with or

The Tilia Alsie viewed from the side, based on photograph.
Illustrates the relatively high positioning of the crew.

0    2 m

Propulsion using paddles: paddling force not constant
since a crew will get tired. Sail generates constant power

WIND

WIND

Port    Starboard

Port    Starboard

With the wind pressing on the starboard side of the boat, the crew on this side
will have to stop or reduce their paddling speed in order for the boat to maintain
an even direction. The crew on the port side will have to paddle with more force
if the boat is to maintain speed and course or heading. If not, the boat will veer
to the left.

When the crew is unable to hold the boat against the wind (due to fatigue), it
will broach, i.e. it will veer further to the left until it lies sideways to the wind. This
will expose its side, the lowest point of the vesse, to the wind threatening the
boat to be swamped with water from waves hitting it from the side.

For this reason the two steering oars are of paramount importance in ensuring
the power generated by the paddlers on each side remains more even and can
be used to propell the boat forwards (meaning starboard side crew might need
to paddle only slightly less and makes it less heavy for the port crew.

Crew on starboard and port side paddling at
even speed and with even force between the
two sides. The boat goes forward and direction
of movement is maintained, aided by the two
steering oars     ).

**Fig 17. The effect of wind and steering oars on a boat being paddled.** One of the biggest problems with the available performance data is that the Tilia Alsie has not been tested in waves other than on one occasion [83]. Again, the test was made in relatively sheltered waters so that even in wind conditions of 20 knots and of gusts of up to 28—32 knots (!), significant wave height encountered was only 1 meter. Provided the boat is made lighter in the bow (with crew and cargo being moved aft), it is estimated that the plough rail and the horn projections should help deflect bow waves of up to c. 2 meters height [74]. However, variability in the distance between gunwale and sea surface in-between each paddle stroke on account of waves rolling past, has a considerable effect on stroke efficiency and as a result crew endurance.

against the waves, rather than at an angle to the waves. This latter might have a large impact on navigational error (compare with navigational error in Fig 9). Thus, any simulation results will have to be checked against these limitations in wave height and wind strength.

## Methods for simulating sea trips using a voyaging modelling tool

Trips between Thy and Lista are simulated with an ocean voyage agent-based model [91, 92] updated here to include the effects of navigational error. Vessel displacement is given by the sum of propulsion and current leeway velocities, with both of those dependent on winds and currents in the proximity of the vessel. Propulsion velocity is a function of the paddling speeds provided by the polar diagrams described above and vessel bearing. Current leeway velocity is the same as the local surface water velocity (Fig 18). The model code can be freely obtained at https://github.com/mtomasini/PLOS-Voyager.

Environmental conditions only affect displacement (the wind resistance of the active crew is allowed for withing the polar diagrams – any non-active crew members are assumed to be seated on the floor)

While strong opposing winds and currents can drastically decrease the rate of progress toward the destination and even impede boats from reaching the end of the trip, vessels do not lose efficiency, become incapacitated or sink. The impact of adverse conditions is evaluated in the analysis of modelled trips. Note that given the potential impact of winds and currents on vessel movement, differences between the direction of vessel displacement and the vessel's bearing toward the destination can exist (Fig 19).

Simulated trips adopt two types of navigation strategies: open ocean and coastal. The following procedure is adopted in open ocean navigation (follow with Fig 19): At the start of the experiment or time 1 (T1), a vessel is positioned at a departure point (P1) and a bearing (B1) calculated based on P1 and the location of a destination point. Environmental information in the vicinity of P1 (dashed blue line) is obtained and displacement (D1) during a "time step" calculated by the model based on locally determined propulsion and current leeway velocities. Time step is the period at which vessel position is updated and kept at 15 minutes for all simulations (open ocean and coastal) discussed here. The displacement calculated at T1 (D1)

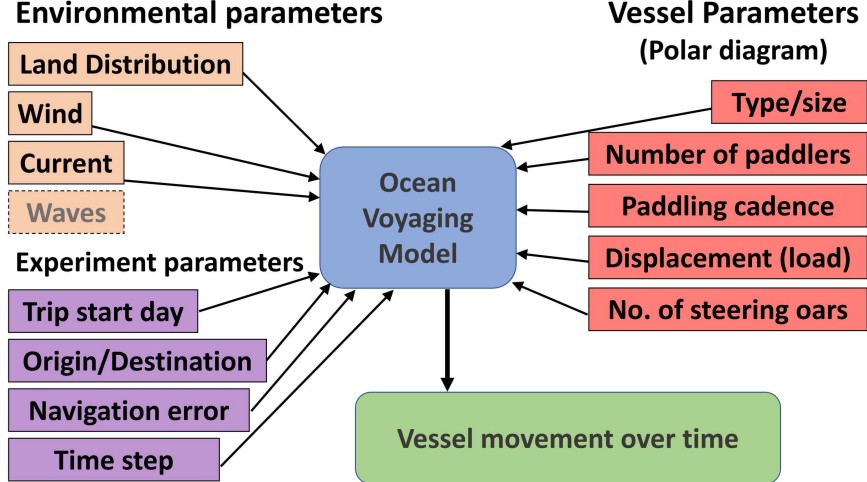

**Fig 18. Voyaging model inputs and output.** Wave information does not influence vessel displacement but is adopted in interpretation of results.

is added to P1, generating a new position, P2, at time T2 (T2 = T1 + 15 minutes). The procedure is repeated until either the vessel arrives at its destination or the end of the experiment – 72 h– is reached. The open ocean navigation strategy is used to simulate direct trips between Thy and Lista across the Skagerrak strait and between the central western Swedish coast and the eastern Danish coast across the Kattegat strait.

The coastal navigation scheme's goal is to make vessels move from origin to destination within a "coastal corridor", so that boats do not approach too close to nor move too far from the shore (follow with Fig 20. See also [91]; appendix 2.2). This is done by having vessels navigate toward a sequence of numbered "provisory targets". Provisory target numbers increase from origin to destination. While origin and destination points are related to locations over land, these are positioned over water within the coastal corridor. At any time step, the model identifies the provisory target closest to the vessel and calculates a bearing toward the next target in the sequence. So that the first bearing (B1) is oriented toward provisory target 2 ($X_2$). As with direct crossings, bearings are defined at each 15-minute time step and depend on vessel and provisory target position. Vessels keep moving along the coast until either destination or end of experiment - 30 days - is reached.

The adoption of sequential targets provides a trustworthy first order representation of coastal displacement but may overestimate trip duration by forcing boats to follow the coast in situations where navigators might be able to sail across the mouth of an embayment instead of following along its coast. Despite this limitation, we believe sequential targets, given the lack of knowledge of Bronze Age navigation strategies and the large difference in duration between open ocean and coastal trips is a viable option given our present goals.

Even with the adoption of sequential provisory targets, strong winds and currents can overcome propulsion speeds and push a vessel against the coast, causing a non-desired landing. To avoid this, when vessels come within 0.05 degrees distance of land (~6 km on the E-W and

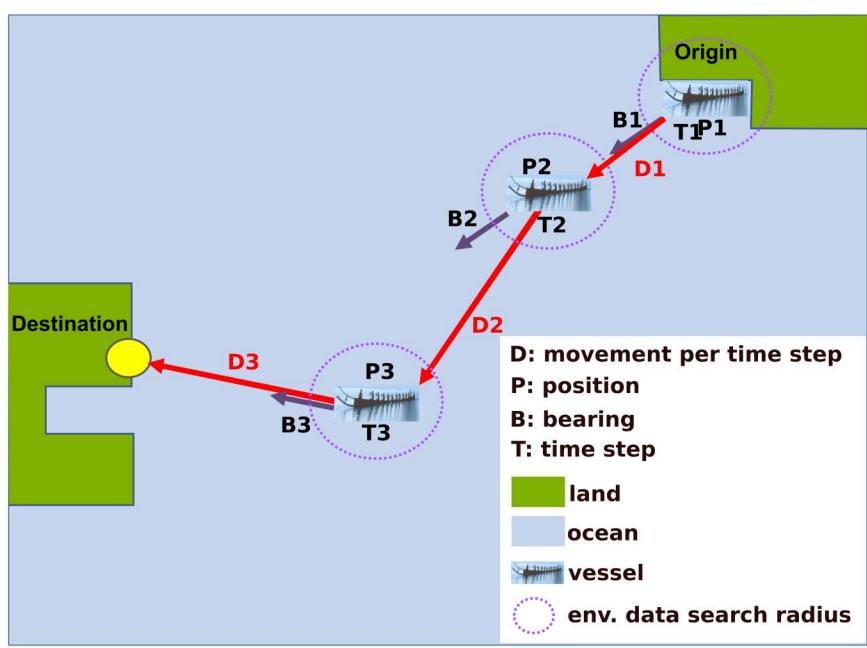

**Fig 19. Simulation procedure for open ocean trips.** See text for detailed explanation and legend for abbreviations. In the example above, the vessel reached its destination in three time steps. The dashed circles around vessels indicate the environmental data used to calculate each displacement is obtained from the vicinity of the vessel.

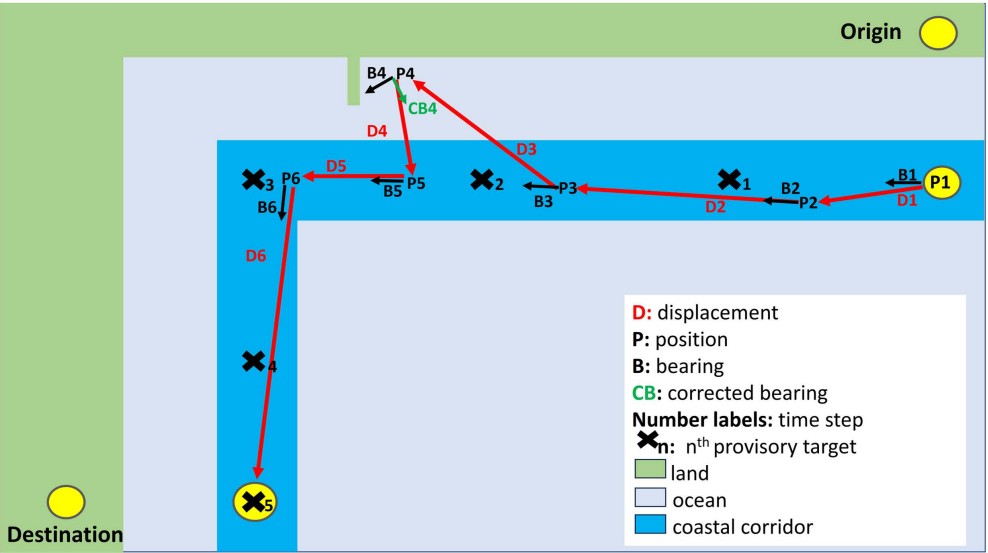

**Fig 20. Simulation procedure for coastal trips, see text for detailed explanation and legend for abbreviations.**
Coastal trips adopt the same environmental data search radius as open ocean experiments (see Fig 19). For clarity, the data search radius and vessel image are not shown here. In the example above, the vessel reached its destination in six time steps.

~ 3 km on N-S directions) the model adopts a "corrected bearing" for the next time step. The corrected bearing is determined by deflecting the original model bearing by 90°. Using Fig 20b as an example: due to the vessel's proximity to land after the displacement calculated for time step 3, bearing B4 is changed to corrected bearing CB4 so that CB4 = B4-90°. The negative sign here causes the vessel to stir away from the land; in case of land present on the left hand of the vessel, a positive deflection angle would be added.

Given the lack of information on seafarers' navigational skills, open ocean crossings are simulated under three distinct scenarios to overcome problems related to navigational error (see Fig 11). In the no error experiments, boats have perfect bearing and are oriented towards the target at all time steps (the equivalent of using modern sat nav data or GPS but might also reflect coastal routes with regular distinct landmarks). In the 10-degree (10°) error experiments, bearings have a random, normally distributed error with standard deviation of +/- 10° from the perfect bearing. This is a probable scenario for most short open (out of sight of land) sea passages in prefect weather and, potentially also for longer open water crossing depending on the skill and experience of the navigator. The bearing error is calculated and changes for each time step. The same procedure is followed for the 25-degree error simulations, with the error ranging from +/-25°, allowing for an exaggerated, worst scenario navigational error. Since seafarers performing coastal voyages can constantly sight land, and assuming this means they would be able to better determine a bearing parallel to the shore, coastal trips are only simulated with perfect and 10-degree error bearings.

The period of daylight is an important factor in both simulations and analysis of results. We choose to adopt as length of daylight the time between the start of civil dawn and end of civil dusk; meaning the period of the day starting when the geometric center of the sun is 6° below the horizon before sunrise and 6° below the horizon after sunset. This is done because during civil twilight the horizon and land features can be discerned and hence navigation deemed feasible. Depending on atmospheric conditions, the horizon might remain visible up to the nautical twilight, when the geometric center of the sun is 12° below the horizon. We

adopt civic twilight as a conservative (shorter daylight period) approach as the potential existence of some illumination at the end of daily simulated displacement is part of the rationale of how coastal trips experiments are designed.

Coastal trip experiments assume boat occupants would avoid travelling at night. Trips start at sunrise from a departure point within the coastal corridor and displacement is calculated until sunset on day 1. At this time movement stops until the sunrise of day 2, and so on until either the destination is reached, or 30 days go by. That is, the last position at sunset of day N provides the initial position at sunrise of day N + 1. A back-and-forth trip between this position and land is implicit but not simulated. The assumption is that the movement to and from the coast would take place under illumination still available between civic and nautical twilights. This means that coastal trip simulations reproduce optimal utilization of daylight. Open ocean trips also start at sunrise but are allowed to continue after sunset for up to 72 h.

Open ocean and coastal trips are simulated for both directions of travel (Thy to Lista and vice versa). Trips are initiated daily for the 28-year period - determined by the availability of environmental data - between January 2, 1993, and December 31, 2020; resulting in 10,226 voyages simulated for each experiment.

## Environmental data

Land distribution, wind and current information comes from present day Era5 reanalysis data generated and distributed by the European Center for Medium Range Weather Forecast (ECMWF) [93]. Reanalysis data can be understood as a blend of modelled and observed values organized on regularly spaced grids and at constant time intervals. These are commonly adopted by voyaging simulations as they provide the temporal and spatial resolutions and coverages required by models (Montenegro, 2024). Original wind data consists of three hourly values with 5.5 x 5.5 km spatial resolution. Original currents consist of daily values with spatial resolution of either 4x4 km or 0.111°x0.067° (or approximately 1.2 km in the north-south direction and 4 km in the east-west direction). Waves do not influence modelled vessel displacement, but hourly wave data with 2x2 km resolution coming from the same ECMWF product is used in analysis of results. Prior to inclusion in the model and analysis, all original environmental data sets are interpolated into a 2x2 km grid with 1 hour temporal resolution.

Tidal flows, which can be significant in channels and near shore areas, are not present in the adopted current data. While tide should be considered when simulating particular trips being performed at specific dates and times, they play a much smaller role in affecting results in the type of analysis conducted here. Our vessels started/ended their displacements without any consideration for tidal phase. Assuming tidal currents were present in our input data, this would mean that among the 10k + voyages simulated per experiment, trips would be initiated/ended under a large range of tidal amplitudes. Some trips would be started/ended under tidal currents in their favour, some with currents against them, other under weak tidal currents with little impact on vessel movement. If we were to average these tide-influenced results into the multi-year daily and monthly means presented below, the most significant tidal effects would be "averaged out". More than that. Tides in the region are mostly semi-diurnal. In both daily coastal trips and open ocean crossings, vessels are in the water for multi-hour periods that last about 1 to almost 2 dominant tidal cycles. This means that, given their durations, even individual trips tend to experience tidal currents that both favour and hinder displacement toward a specific direction. In short, given how model output is treated and mean individual trip duration, the adoption of input currents that contained tidal flows would not have had a large impact on our conclusions.

The use of present-day environmental information as a proxy for earlier climate states is a common strategy when it comes to ocean voyaging simulations, particularly for those dealing with events taking place over the last 5 thousand years, when at least the global climate has not changed significantly [94–97]. The alternative would be either paleoproxy-based local reconstructions or output from paleoclimate model simulations. Although significant uncertainty exists in wind and current reconstructions, it could be argued that paleoproxies could provide more representative values at a particular location. However, these do not have anywhere near the spatial coverage and temporal resolution required by the voyaging simulations. While paleoclimate models solve the spatial coverage problem, most of these have spatial resolution of many tens to hundreds of kilometers and we are not aware of available paleoclimate model output with the spatial resolution required by our experiments. Another limitation is that, as with all global climate models, paleoclimate simulations tend to underestimate environmental temporal variability [98, 99]. Given the importance of seasonal variability in the region and the short duration (order of hours) of our simulated trips, we believe that present day reanalysis products offer, at this time, the best available input data for our experiments.

## Results

Analyses of the results are focused on generating quantitative estimates of risk and effort involved in trips between Thy and Lista and on describing how these changes seasonally. Trip risk and effort are evaluated in terms of trip duration - particularly in relation to availability of daylight - waves, winds, and currents encountered by boat occupants during trips. Daily and monthly values represent averages covering the 28 simulated years.

### Thy-lista open ocean trips

Simulated trips that arrive at their destinations tend to display relatively direct trajectories. As expected, trajectories are more direct (straighter) when smaller navigation error is adopted. During the April to August period more amenable to voyaging, no clear variation of trajectory shape as a function of trip direction or time of year is observed (Fig 21).

Even when averaged over 28 years, there is marked day to day variability on environmental conditions faced by travelers (Fig 22) and trip duration (Fig 23). No matter the direction (Thy to Lista or vice versa), about 10% to 80% of trips encounter wave heights below or equal to (≤) 1 m, with lower percentages taking place in winter and higher values between April and August (Fig 22). Trips can be finished under daylight in both directions between March (April) and September (August) for a 10º (25º) navigational error. During the optimal late spring to summer months, a larger percentage of southern trips (~20% to 65%) finish under daylight compared to northward voyages (~10% to 50%). The period when both daylight and wave conditions are met continuously range from about April (May) to August (July) for the 10º (25º) navigational error. During this interval the percentage of trips under both thresholds vary from about 10% to 30%.

When only the wave threshold is considered, average trip duration tends to be more than 20 h, with many trips, particularly southward ones, lasting 30 h or more (Fig 23). If daylight and wave limits are considered, spring and late summer trips tend to be shorter than summer ones, with larger navigational error leading to longer trip times. In general terms, between late spring and late summer, trips in both directions that last about 16 h to 19 h, are finished under daylight and encounter average waves heights below 1 m can take place at a rate of about once a week (Figs 23 and 24).

Daylight and wave height limits identify risks associated with navigation, such as becoming lost at sea or stranded; and vessel stability, like capsizing or sinking. Wind intensity also

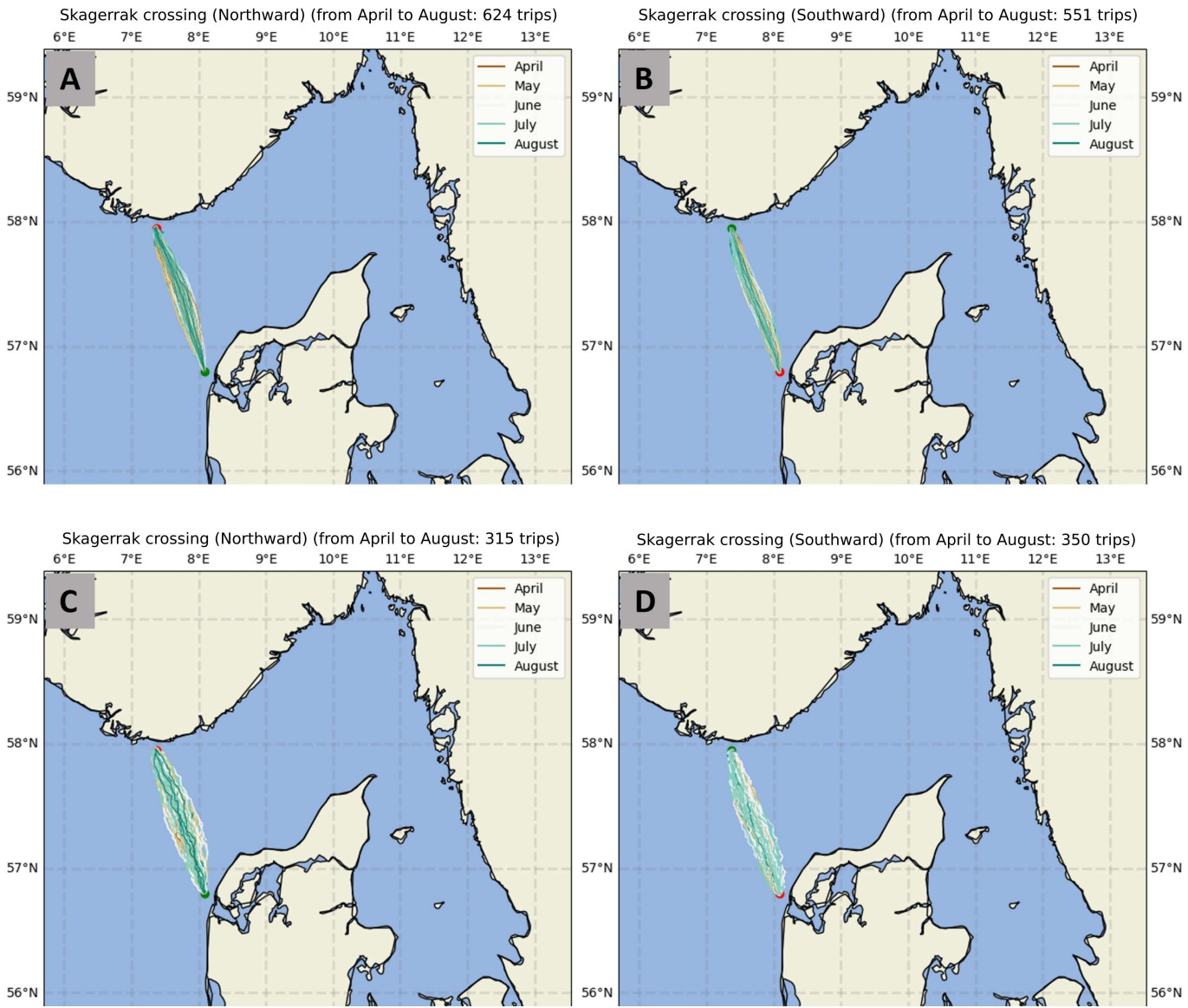

**Fig 21. Successful trajectories of: A and C, northward trips with 10° and 25° navigational error respectively; B and D, same as A and C for southward trips.** Colours refer to trip month. Maps generated using the Python package Cartopy (v0.24.1. 09-Oct-2024. Met Office. https://github.com/SciTools/cartopy/releases/tag/v0.24.1). Cartopy uses public domain data from Natural Earth.

constitutes a risk, not only via generation of sea waves, but also by potentially generating leeway velocities that would require large paddling efforts to be compensated, efforts that might not be sustainable for the many-hour trips between Thy and Lista. To evaluate the joined effects of winds, waves and day length, the polar diagram (see above) and expert opinion based on sea trials of the *Tilia Alsie* are used to identify what here we define as low-risk and ideal open ocean voyaging conditions. Low-risk trips are defined as trips finished under daylight and encountering average winds speeds and wave heights no greater than 10 kt and 1 m respectively. Ideal trips also finish during daylight with wind speed and wave height thresholds of 6 kt and 0.4 m.

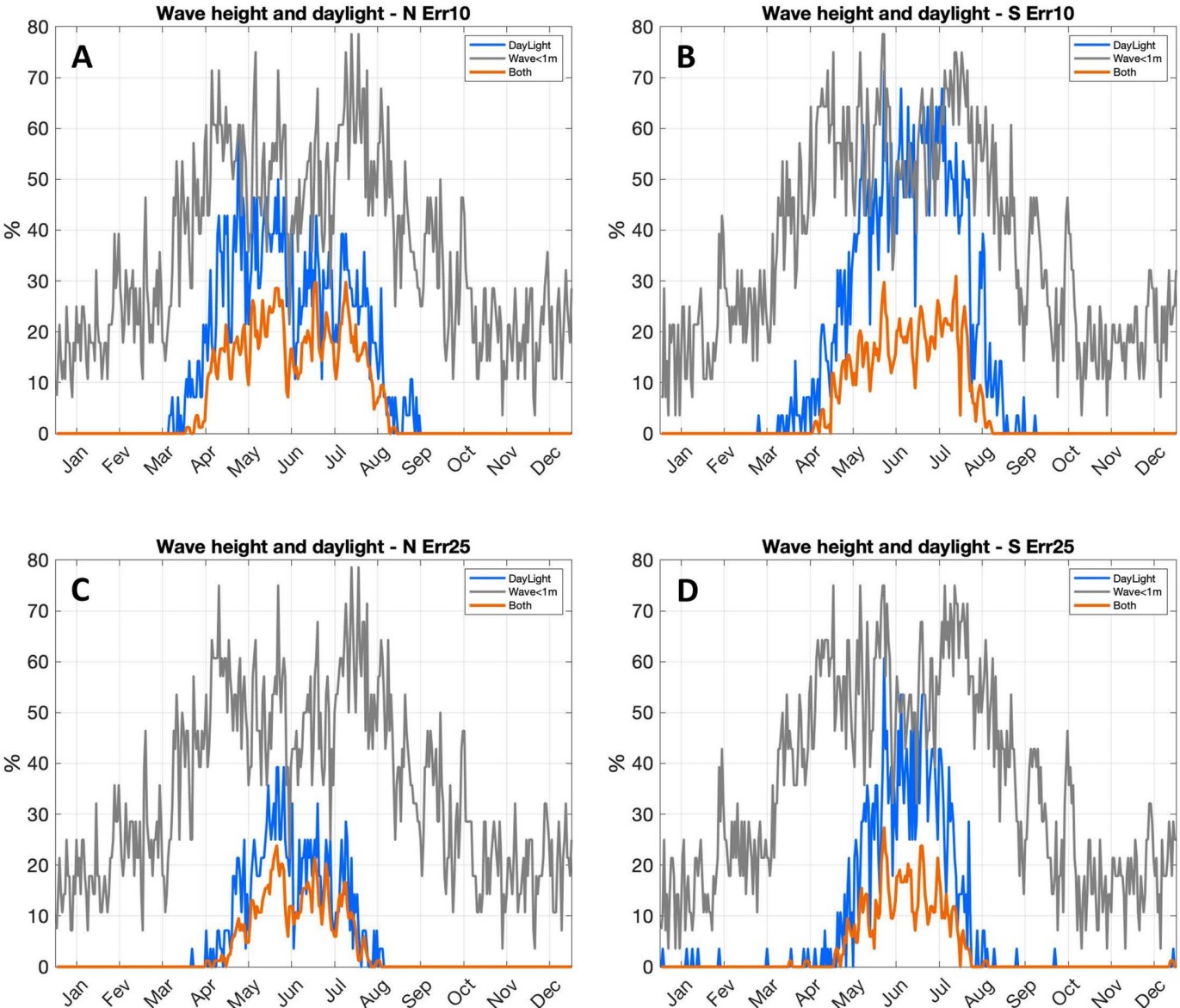

**Fig 22. Average daily percentage of trips completed under waves heights of 1 m or below (grey); under daylight (blue) and when both previous conditions are met (orange).** A, northward trips (Thy to Lista) with 10° navigational error; C, same as A for 25° error. B, southward trips (Lista to Thy) with 10° navigational error; D, same as B for 25° error.

Low-risk trips in both directions take place from April and August; between May and July about 6% to 21% trips occur under low-risk conditions, with higher (lower) value related to the 10° (25°) navigational error results (Fig 24, Table 2). Trips taking place under ideal conditions are restricted to the period between May and August. They are much less common than low-risk ones, with the highest frequencies (~1.3% to 2.6%) seen in July for both directions (Fig 24, Table 2). Differences in navigational error do not have a large impact on the percentage of trips taking place under ideal conditions. This analysis points to June and July as the best months for travelling between Thy and Lista, with low-risk trips possible every five to seven days and ideal voyaging conditions occurring every 36 to 70 days, depending on

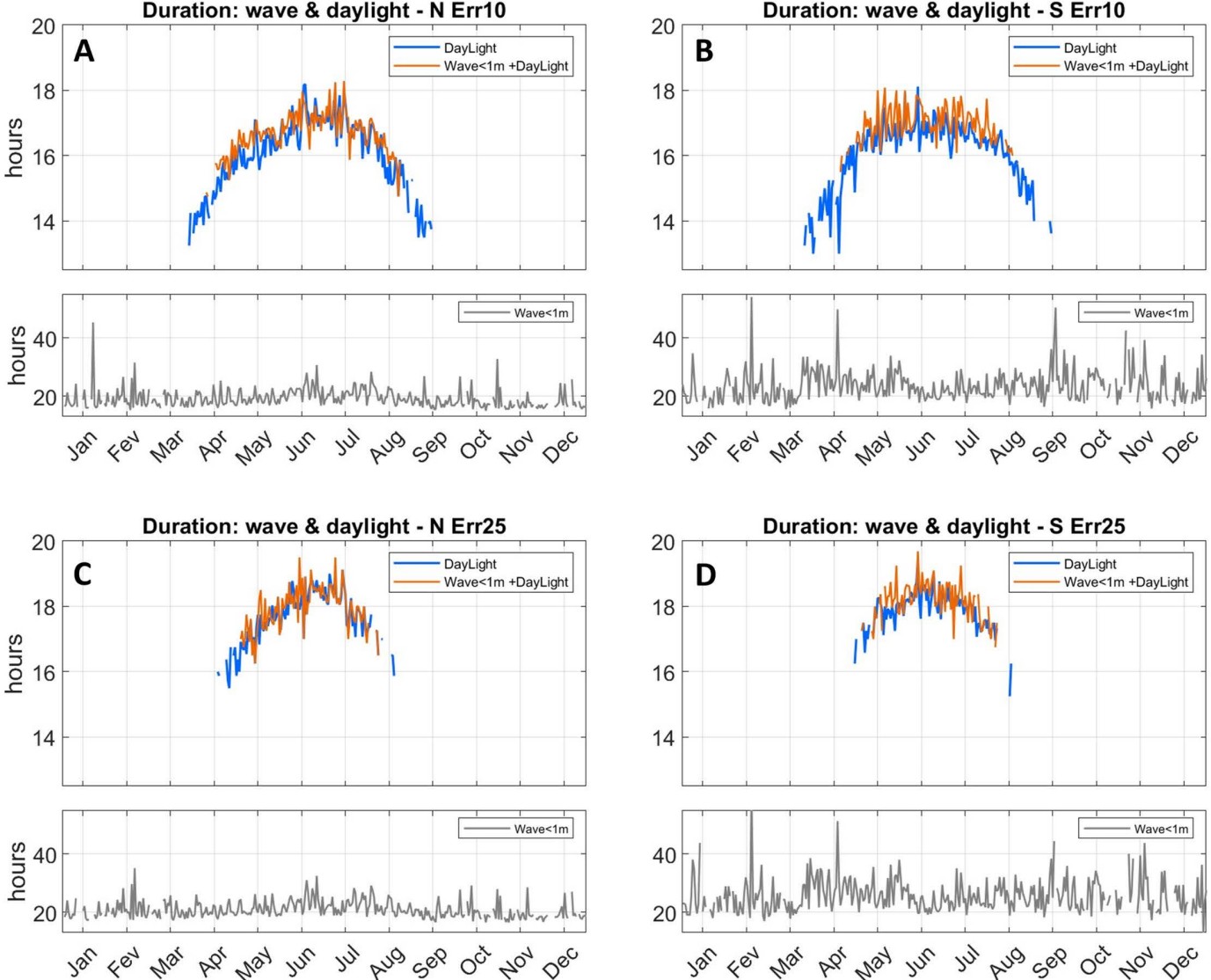

**Fig 23. Average daily duration of trips completed under waves heights of 1 m or below (grey); under daylight (blue) and when both previous conditions are met (orange).** A, northward trips (Thy to Lista) with 10˚ navigational error; C, same as A for 25° error. B, southward trips (Lista to Thy) with 10˚ navigational error; D, same as B for 25° error. Wave heigh limited trips have much longer duration and are plotted separately.

the assumed navigational error (Fig 24, Table 2). Ideal and low risk trips tend to take place with winds and currents more or less aligned with the desired bearing, although portions of the trips can take place with environmental flows at large angles or even against the desired direction of travel (Fig 25).

## Thy–Lista coastal trips

For coastal voyages, the portion of the trip performed along the Norwegian and Swedish coasts is analyzed separately from the open ocean crossing of the Kattegat. Trips along the shore from Lista to the eastern margin of the Kattegat (Fig 26) last from about 11 to 20 days, with longer travel times from September to February and faster trips between April and August. The same seasonal pattern is simulated for movement in the opposite direction, but trips from the

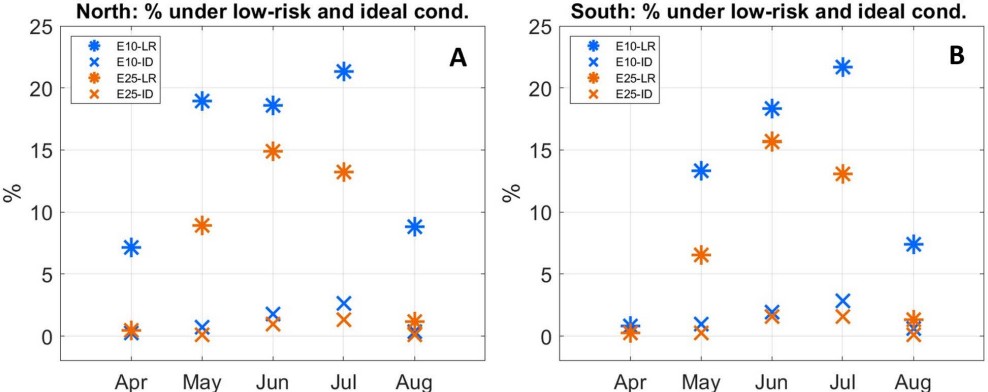

**Fig 24. Percentage of low risk (LR, asterixis) and ideal (ID, Xs) conditions for the months in which these take place.** Blue and orange refer to navigational errors of 10 and 25 respectively. Left (A) and right (B) panels refer to northward and southward trips respectively. Low risk is defined as trips finished under daylight and encountering average winds speeds and wave heights no greater than 5 m/s (10 kt) and 1 m respectively. Ideal trips also finish during daylight with wind speed and wave height thresholds of 3 m/s (6 kt) and 0.4 m. Values are monthly means.

**Table 2. Percentage (% Trips) and average duration (Ave Dur), in hours, of trips finished under low risk (L Risk) and ideal (Ideal) conditions for the months in which these take place. Low risk is defined as trips finished under daylight and encountering average winds speeds and wave heights no greater than 10 kt and 1 m respectively. Ideal trips also finish during daylight with wind speed and wave height thresholds of 6 kt and 0.4 m. North and south refer to trip direction. E 10 and E 25 refer to navigation errors of 10 and 25 respectively.**

|  |  | Apr | | May | | Jun | | Jul | | Aug | | Average | |
|---|---|---|---|---|---|---|---|---|---|---|---|---|---|
| **North E 10** | | L Risk | Ideal | L Risk | Ideal | L Risk | Ideal | L Risk | Ideal | L Risk | Ideal | L Risk | Ideal |
| | **%** | 7.1% | 0.2% | 18.9% | 0.7% | 18.6% | 1.8% | 21.3% | 2.6% | 8.8% | 0.4% | 15.0% | 1.1% |
| | **Dur** | 15.8 | 16.4 | 16.6 | 17.2 | 17.1 | 17.7 | 17.1 | 17.5 | 16.4 | 16.3 | 16.6 | 17.0 |
| **North E 25** | | L Risk | Ideal | L Risk | Ideal | L Risk | Ideal | L Risk | Ideal | L Risk | Ideal | L Risk | Ideal |
| | **%** | 0.5% | 0.0% | 8.9% | 0.1% | 14.9% | 1.0% | 13.2% | 1.3% | 1.2% | 0.1% | 7.7% | 0.5% |
| | **Dur** | 16.5 | na | 17.5 | 17.8 | 18.3 | 18.8 | 18.1 | 18.6 | 17.2 | 17.5 | 17.5 | 18.2 |
| **South E 10** | | L Risk | Ideal | L Risk | Ideal | L Risk | Ideal | L Risk | Ideal | L Risk | Ideal | L Risk | Ideal |
| | **%** | 0.8% | 0.0% | 13.3% | 1.0% | 18.3% | 1.9% | 21.7% | 2.9% | 7.4% | 0.6% | 12.3% | 1.3% |
| | **Dur** | 16.1 | na | 17.0 | 17.8 | 17.1 | 17.7 | 17.1 | 17.6 | 16.5 | 16.8 | 16.8 | 17.5 |
| **South E 25** | | L Risk | Ideal | L Risk | Ideal | L Risk | Ideal | L Risk | Ideal | L Risk | Ideal | L Risk | Ideal |
| | **%** | 0.2% | 0.0% | 6.6% | 0.2% | 15.7% | 1.6% | 13.1% | 1.6% | 1.3% | 0.1% | 7.4% | 0.7% |
| | **Dur** | 16.1 | na | 18 | 18.8 | 18.5 | 19 | 18.2 | 18.7 | 15.8 | 18.5 | 17.3 | 18.8 |

Kattegat to Lista are slightly faster, with durations ranging from about 9 to 17 days (Figs 26 and 27). As seen in the open ocean crossings described above, there is significant variability on trip duration depending on departure day. These durations should be interpreted as minimum voyaging times, or as number of required "days on the water", as they refer to continuous traveling with no rest periods other than the nightly stops (see Methods). Considering wave risks, coastal trips in both directions are safer between April and August, when they do not encounter maximum wave heights above 2 m (Fig 27). The slightly higher maximum wave heights faced by southward trips is a function of their longer duration. Simulated vessels encounter lower waves along the coast compared to the open ocean, but given the adopted data set spatial resolution, even those lower coastal wave height values are very likely overestimates of what conditions would be like well protected areas between the shore and near shore islands.

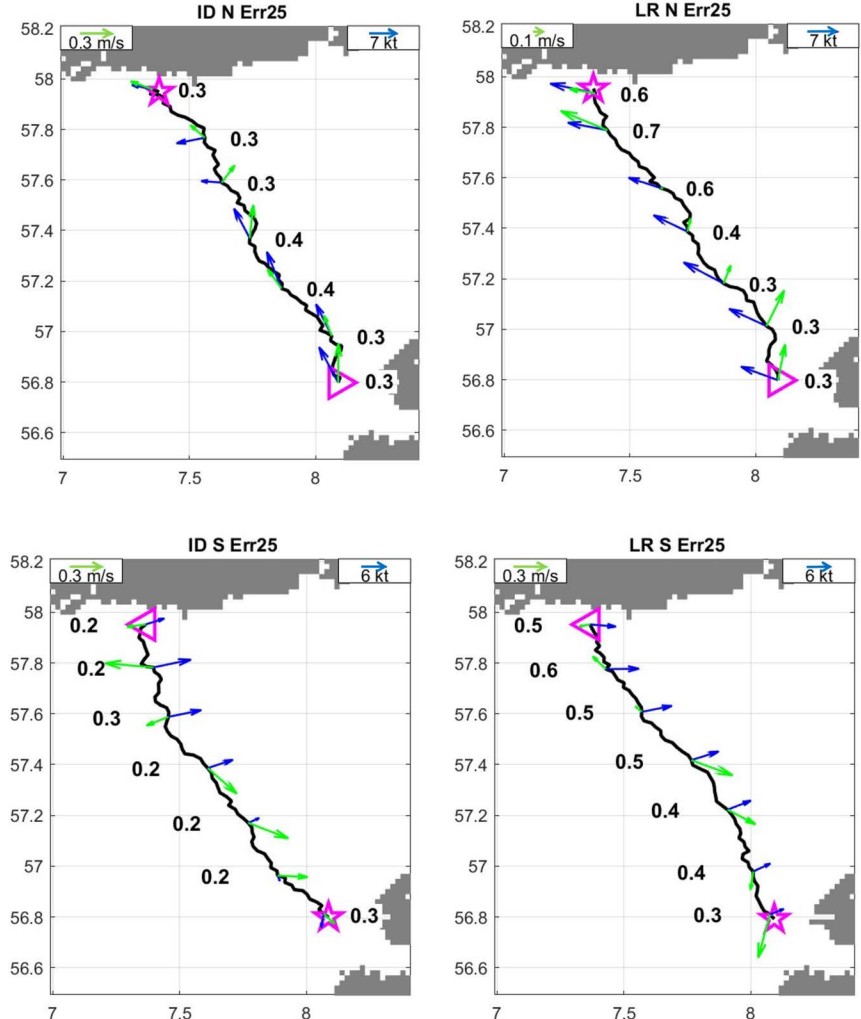

**Fig 25. Example of trips under ideal (ID, left column) and low risk (LR right column) conditions.** Top and bottom figures come from northward (N) and southward (S) trips respectively. All trips have 25 navigational error (Err25). Black line, trajectory; black numbers, wave height; blue and green vectors, wind (in kt) and current (in m/s) velocities, respectively. Magenta triangle and star mark start and end of trip.

As expected, given the shorter distance and more protected ocean environment, crossing the Kattegat is less risky and much faster than the open ocean voyages between Thy and Lista (Fig 26). The fact durations range from about 7 to 11 hours allow crossings under daylight even during the shortest winter days; and even between December and January, sheltered conditions mean that trips under daylight and encountering waves no higher than 1 m can occur about once every 5 to 6 days, or ~ 20% of the time (Fig 27). April to August provide the highest frequency of safe travel days, with daylight, low wave height trips possible for about 50% to 75% of the time. During this safer period for travel, trips last about 8 h to 11 h, and trips from Denmark to Sweden are slightly faster than those in the opposite direction (Fig 28).

## Discussion

The ocean modelling tool presented in this paper uses modern environmental data and predicted performance data of a BA type vessel (the latter, compared to on-water performance

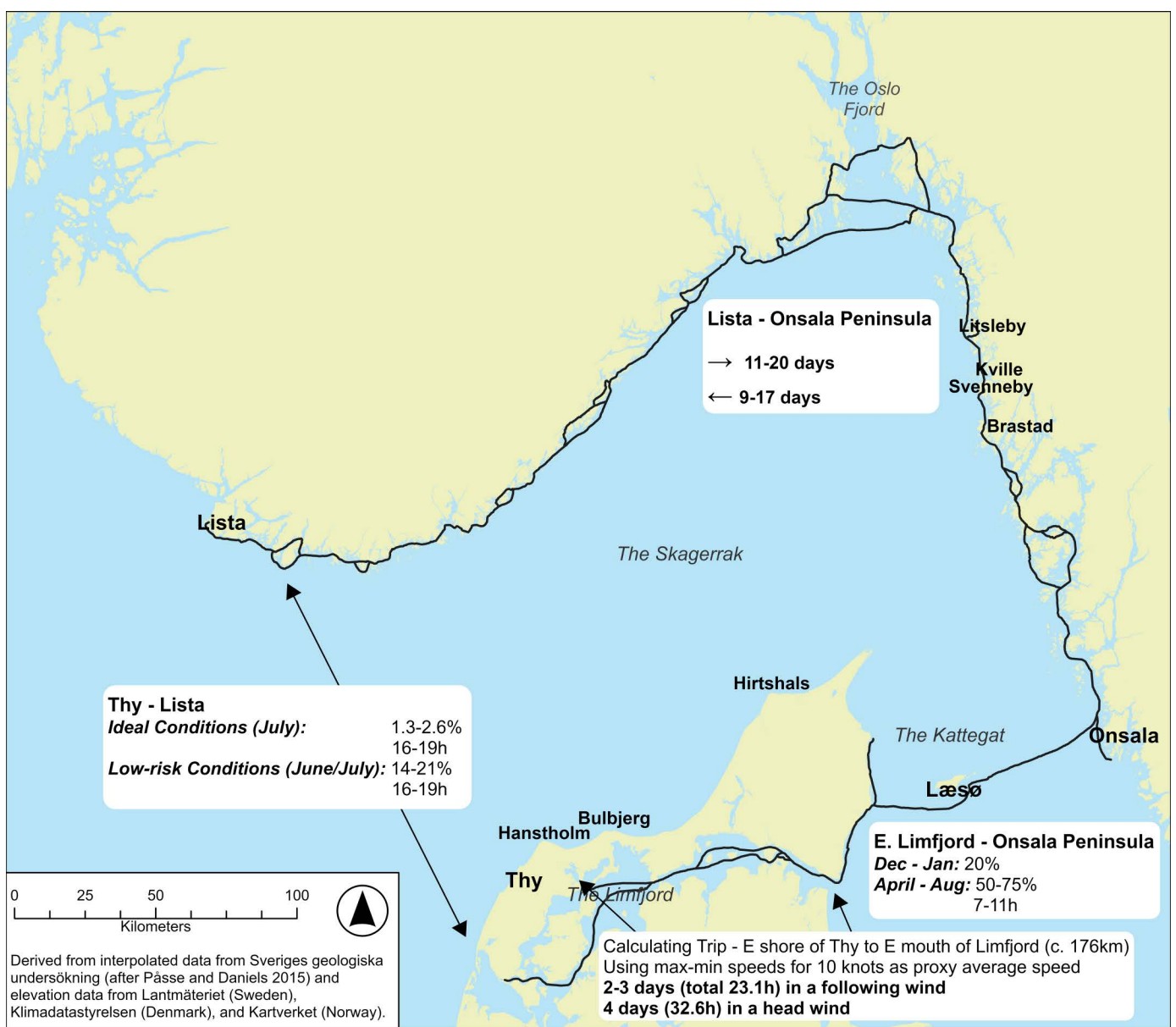

**Fig 26. Synthesis of results.** Simulated coastal trips from the Onsala Peninsula region to Lista takes on average 9 to 17 days (in a counterclockwise direction) with the quicker times from April to August, and with trips in the opposite direction on average two days longer. Top two boxes provide percentage of successful simulated trips and their minimum average duration in hours. Values for the trip along Limfjord (bottom box) are based on average propulsion and wind speeds and not on simulations. calculations alone. Based on best case scenario travel time between Thy and Lista varies between c. 19 hours (direct crossing of the Skagerrak) to c. 13 days minimum when following the longer coastal route. This map shows a paleoDEM created by the authors in ArcGIS Pro using elevation data from Lantmäteriet, Klimadatastyrelsen, and Kartverket under a CC-BY license and shore displacement values provided by SGU (after Påsse and Daniels 2015). Note: None of the original elevation data is displayed.

data) to discuss communication and potential trade between Thy and Lista in the Early Bronze Age of Scandinavia. The close similarities within the archaeological record for these two regions and their relative geographic situation on either side of the Skagerrak has invited suggestions of direct open water sea crossings of 110 to 180 kilometres already at the onset of the Early Bronze Age (i.e., Late Neolithic), which, depending on the weather conditions and visibility, might have included up to 50 kilometres of navigation out of sight of land. The only

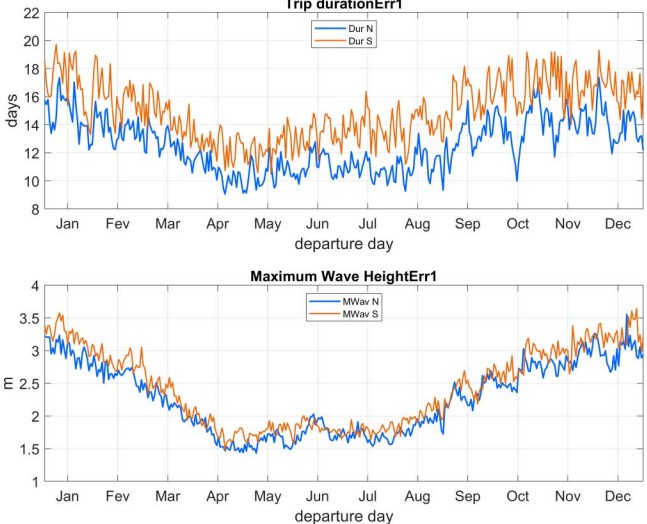

**Fig 27. Top, daily averaged duration in days for coastal trips between Lista and the eastern margin of the Kattegat (Dur S, orange) and in the opposite direction (Dur N, blue) for trips departing at different days of the year.** Bottom, same as top the but for maximum wave height encountered during the whole trip. All results for trips with 10-degree navigational error.

other viable alternative for explaining the close similarities within the archaeological record in the two regions is that communication mainly took place along the 700 km long coastal route, which we suggest might have involved following the Limfjord to the east from Thy, crossing the Kattegat via the sheltered waters surrounding present-day Læsø and proceeding onwards within the shielded archipelagos [20].

Whilst the direct Skagerrak crossing might have taken 16-19 hours, to which time to the simulated points of departure/arrival from any nearby landing sites that might represent the actual points of departure/arrival (the simulations must start from a bit further out at sea due to availability of environmental data) should be added, the longer route would take between a minimum of 12 to 22 days (Fig 26). Also, for this longer route additional time most likely must be added but for different reasons (see below).

Any uncertainties in the predicted performance of an Early BA type vessel based on the c. 350 BC Hjortspring boat, navigational abilities, and questions related to by what extent today's environmental conditions agrees with prehistoric conditions, have been addressed during the process of setting parameters and limitations for these simulations. Importantly, any of these parameters and limitations can be further altered and adjusted in future simulations.

Based on the assumption that navigation was primarily undertaken during hours of light (with the exception of longer open sea voyages during months when the sun never sets beyond 8 degrees below the horizon), these simulations suggest attempts to paddle across the Skagerrak in a BA type boat would have been limited to the months of April to August if allowing for a larger (exaggerated) navigational error and, when applying wave data, further limited to between May and July. Furthermore, it is clear that ideal environmental conditions (winds of ≤ 6 knots and waves ≤ 0.4 m), representing the conditions the Hjortspring boat is ideally suited for, only occur for 1.3-2.6% of all simulated trips during these months (one day every 30-70 days or the equivalent to c. 1-3 times per seafaring season) which does not satisfactorily explain the archaeological record and would further suggest an ability to pick out these particular days to be of use for early seafarers intent on this passage. Given the large

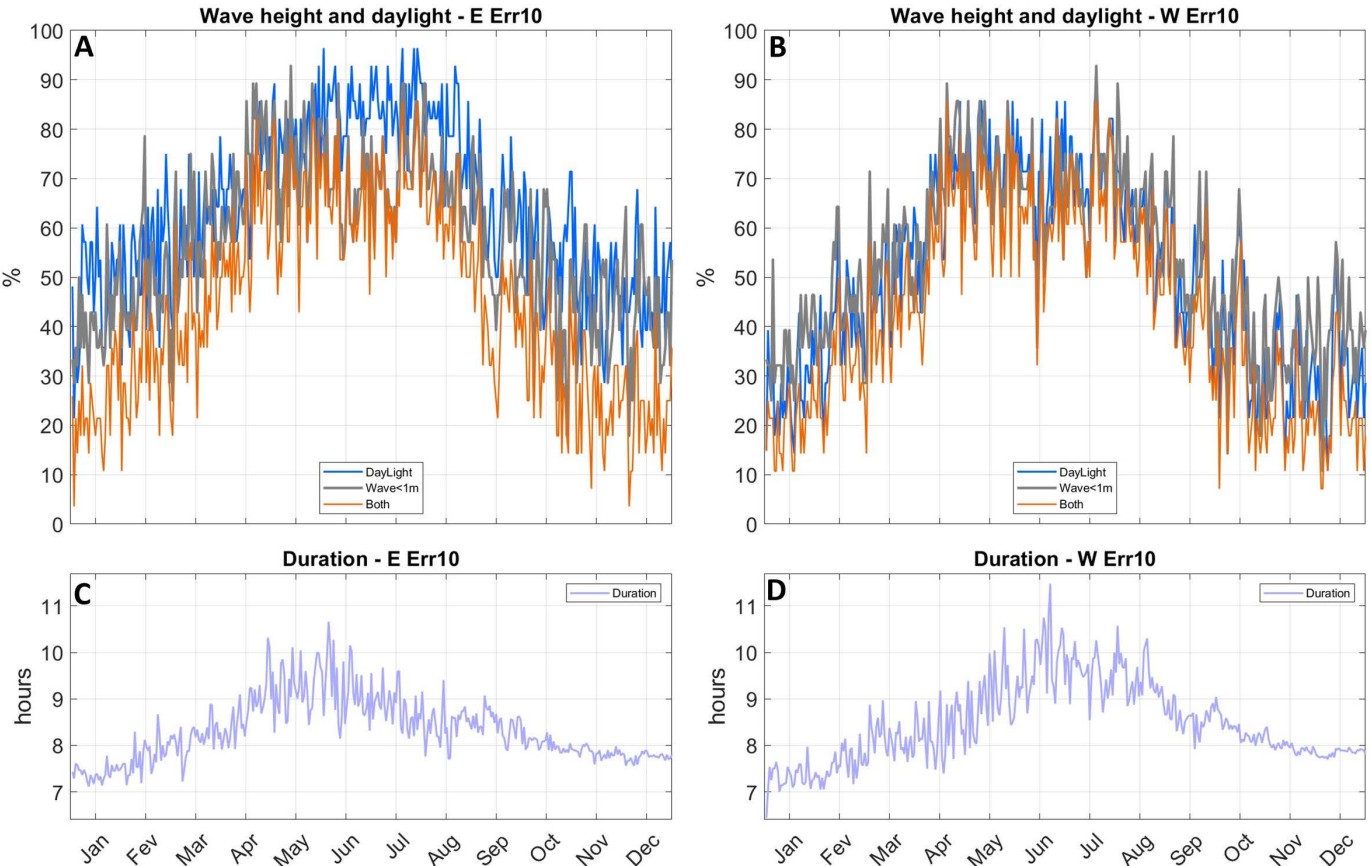

**Fig 28. A daily average percentage of trips completed under waves heights of 1 m or below (grey); under daylight (blue) and when both previous conditions are met (orange) for trips crossing the Kattegat from Denmark to Sweden; B, same as A, but for trips in the opposite direction; C, duration in hours of crossings from Denmark to Sweden that took place under both wave and daylight thresholds (orange trips in A); D, same as C for crossings in the opposite direction.**

variability in the regional weather, both navigational skills as well as weather predictions skills must therefore have been well developed before these crossings were attempted in the type of boat and propulsion chosen for these simulations. If, therefore, assuming better navigational abilities (within ± 10 degrees) and weather prediction skills to match, while also extending the allowance for the environmental conditions encountered during the direct open water trips to also include low-risk conditions (winds ≤ 10 knots and waves ≤ 1 m), the number of days during which trips might succeed increases dramatically to one in every 5-6 days or roughly one every week during the same period of time. This by itself indicates the dramatic difference in success rate of the open sea crossing simulated here if a vessel can handle larger waves.

The potential skill involved in identifying suitable conditions for a successful crossing is revealed by zooming in on individual simulated trips, a method that can also be used to check the validity of any simulated trips (Fig 23). Here we find that even in ideal conditions, successful trips mainly occur when wind, wind induced waves and currents run in the same general direction as the intended direction of travel. The more wind and the larger the waves the more obvious this relationship becomes. It also means that potentially, provided the boat can be loaded in a way that reduces weight in the bow, and a method of securing the forward steering oar which does not require active steering in this end, one could assume that the BA type vessel used here might have been capable of using a following wind to paddle across also

in a higher wind register provided waves stay within the one meter mark. This latter would of course be very dependent on paddlers capable of maintaining speed for the duration of the trip (compare Fig 17).

Furthermore, the large variability in weather conditions within the region highlights the advantage of speed when attempting a direct sea crossing, as it minimizes the danger of any sudden weather changes. It is also clear that the two large currents in the region (the Jutland stream running along the southern shores of the Skagerrak) and the Baltic stream (running along eastern shores of the Kattegat, and the eastern and northern shores of the Skagerrak) have negligible impact on simulated trips in comparison to the speed and direction of the wind.

The longer coast hugging sea journey simulated here is not as dependent on a particular season, mainly on account of the more sheltered waterways with associated lower wave amplitude. This is evident even though, given the spatial resolution of the available wave data, the wave amplitudes used for simulating the coastal trips do not include the full buffering effect of the archipelago (compare with Fig 11 and Table 1). This also means that a crew with a basic level of seamanship is very unlikely to come into difficulties within the shielded buffer zones. Thus, variations in navigational skills mainly affect the time it takes to reach a destination but has less impact on safety. The length of passage each day is however dependent on available light since the archipelago is difficult to navigate at night unless there is a clear sky and full moon (which we cannot simulate for).

It is important to bear in mind that the simulated results are based on non-stop travel on the assumption that although stop over points during hours of darkness provides additional resting time for the crew, it excludes any time that might be needed to restock food and water. If assuming the boat needs to restock every two to three days, any sea route taken needs to be more carefully planned to include places where food and water can be obtained, which also adds extra days or half-days to the total journey.

In a scenario where a crew paddles non-stop for 19 or 20 hours between two points where they are expected and welcome, extra supplies on-board might be kept at a minimum. The potential danger of dehydration for an active paddling crew, however, means water is more dangerous to cut back on. If assuming the physical exertion of paddling is equivalent to a light jog, a fit male (assuming the average crew is male) who is c. 1.70-1.74 m tall [100], weighs around 65 kg and is between 18 and 30 years of age, might need up to 6 litres of water for a trip of this magnitude to compensate for water loss due to sweating (which is higher the fitter the person is) and consumption of dry food. Fig 29 provides an estimate of the breakdown of the displacement and essential cargo of the paddled BA type vessel used for these simulations, which clearly demonstrates the relatively small amount of extra cargo that can be taken on-board and the level of planning involved in preparing and executing long distance sea journeys, weighing aspects of safety, equipment and the welfare of the crew against the capability of carrying extra cargo [20].

The disadvantage of a relatively small hold becomes even more noticeable when adjusted for the longer coastal route (Fig 26) where speed (number of active paddlers) must be set against the extra amount of cargo that can be taken on-board (roughly 100 kg per paddler). For sea journeys potentially lasting several weeks, a lack of food would become a problem if stocks were not maintained. Not only does the constant paddling require an intake of extra calories, but the body also burns more calories just by being on water (perhaps as much as 40%). Whereas easily digestible dry food that is not easily ruined by salt water might be brought along and consumed en route, a warm meal and a fire might have been desirable every three days or so, not least to keep up morale, and could be timed to coincide with the regular restocking of food and water. This suggests that for the longer journey additional

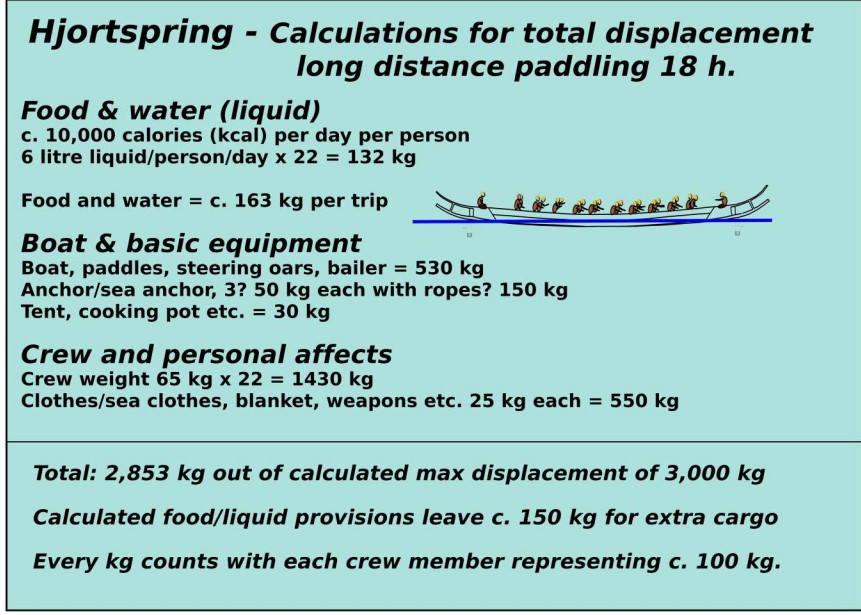

**Fig 29. An estimate of itemised cargo in relation to a total max displacement of c.** 3000 kg equivalent to a fully loaded Hjortspring boat [83], use here to represent a BA type vessel. We assume a full crew would have consisted of 16 active paddlers, an extra four taking turns, as well as two crew members on the two steering oars.

re-stocking time is added, at a minimum, every three days depending on how many days' worth of food/water is carried on-board. As can be seen in the calculations for total displacement based on long distance paddling for up to 10 hours a day (Fig 30), the amount of additional cargo that can be carried to the destination is very limited when propelled by paddling.

The simulations presented here suggest the sheltered coastal waterways can be used all year around and that they therefore are very likely to have served as a form of prehistoric "highway", vastly facilitating sea journeys that could be undertaken within the light hours of a day. It also demonstrates the relative ease of communication within the Limfjord (two to four paddling days from end to end) and the potential for regular communication from the Limfjord across to the eastern shores of the Kattegat, which is possible all year around over the course of a day, but most favourable during the months of April to August when successful crossings can be made every second day to every three out of four days (Fig 26).

The route that best explains the level of similarities within the archaeological contexts in Thy and Lista is the direct sea crossing within the "low-risk" register, which would place significant importance on seafaring abilities, including navigational abilities out of sight of land and short-term weather forecasting, and would imply the use of boats capable of maintaining speed also in waves. While a boat such as the Hjortspring boat would most likely have been capable of making successful crossings under ideal and low-risk conditions in a following wind (using two steering oars) the question of how much cargo that can be carried across and under what circumstances the potential risks involved in such a crossing becomes worthwhile remains at large. It has been suggested an average cargo of raw bronze metals might lie in the region of 80 kg [3], which no doubt any single crossing would be able to accommodate and thus justify the risk. However, this by itself would not explain the close relationship between the Thy and Lista, nor the risks taken by each attempted crossing, Therefore, trades in other commodities as well as alliances must be at play which might include more bulky types of

**Hjortspring - Calculations for total displacement long distance paddling 8-10 h/d.**

**Food & water**

5000-8000 calories (kcal) per day per person

4.5 litre liquid/person/day x 22 = 99 kg

**Food and liquid = c. 120 kg per day**
**Boat & basic equipment (710 kg)**

**Crew and personal affects (1,980 kg)**

Crew of 22 = 1430 kg

Clothes/sea clothes, blanket, weapon 25 kg each = 550 kg

**Total: 2,853 kg out of max 3,000 kg total displacement**

**1 day of food provisions leave c. 150 kg for extra cargo**

**2 days of provisions leave c. 20 kg for extra cargo...**

**Fig 30. An estimate of itemised cargo in relation to a total max displacement of c.** 3000 kg of the BA type vessel used in our simulation, based on 20 paddlers (16 active plus 4 taking turns paddling/bailing) and two on the steering oars during a long-distance trip.

trade including, e.g., slaves, furs and even animals, which would also suggest larger vessel might have been used.

The significance of both the direction of the wind and wind induced waves for the progress and speed of a BA type boat in combination with the variable weather conditions in these waters suggest planning of routes would always have featured in any kind of sea journey. Indirect evidence from within the region now suggest relatively large vessels, as well as sail might have been used already by 1600 BC [3]. The results of the simulations presented here would support this finding on two important points. The first is that even when paddling, crossing the Skagerrak would have been dependent on the direction of the wind [20]. Secondly, bulky cargo would necessitate a reduced crew on a Hjortspring type boat, or alternatively a wider vessel in comparison to its overall length, both of which suggest the additional use of sail in order to ensure success in a direct crossing [3]. This would also better fit with the close archaeological connection between the two regions.

Interesting follow-on questions here include what might have led to the establishment of a direct route between Thy and Lista and when such voyages might have commenced? Here a knowledge of the significance of thermal cumulus clouds forming over land is likely to have been crucial (Fig. 5). Such clouds can reach heights of up to 2000 meters or more when fully developed towards the late afternoon, usually between 5 to 20 kilometres inland from the coast (Fig. 31). During the right atmospheric conditions, such clouds forming over the Norwegian coast are therefore likely to be visible when looking north from Hanstholm, even when calculating on a cloud height of only 1000 meters, and when standing on the beach if such clouds were fully developed. This would have provided the firm knowledge and the general direction of a distant landmass across the Skagerrak, meaning direct crossings might have been initiated from either direction (Thy or Lista). Therefore, direct journeys might have been commenced as and when seafarers were capable of navigating the full distance regardless

### Distances seen from the coast from c. 60 m height

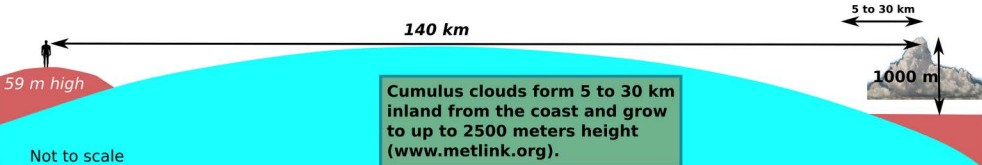

**Fig 31. Showing the distance at which cumulus clouds can be detected during ideal atmospheric conditions.** Calculations here are based on cloud formation reaching a height of 1000 meters. Cumulus clouds can build up to over 2000 meters height and would then have been visible by a person standing on the shore looking across without the need to stand at a height.

of the presence of thermal clouds (or optimal atmospheric conditions), and when the needs for such journeys outweighed the perceived risks of undertaking them. The archaeological evidence suggests this happened already around 2300 BC. At this time archaeological evidence also suggest the northward coast from Lista towards the Stavanger fjord was also regularly used, a comparatively exposed stretch of coast that is likely to have necessitated vessels of similar capabilities as those used for direct crossings of the Skagerrak [3].

Thus, the methods and tools presented here for evaluating prehistoric sea travels also allow for very fine-grained discussions on what the minimum boat requirements would have been for particular sea journeys.

## Conclusions

The significance of maritime travel during the Nordic Bronze Age cannot be overstated, as evidenced by the extensive research published over the last two decades [34,35,101,102]. Although sturdy, these studies tend to present maritime travels as hypotheses based on a combination of theoretical frameworks and archaeological data. With this paper, we have set out to methodologically scrutinise and test these hypotheses with a new ocean voyage simulation tool and the application of boat performance data of a BA type boat. Our findings show that systematic and regular crossings over long stretches of open sea where possible in vessels similar to the Hjortspring boat and that such journeys might have begun already by c. 2300 BC. Nevertheless, successful journeys needed crews with extensive maritime know-how, and seafaring skills. Even though the longer and safer journey between Thy and Lista would seem like the most obvious choice, we argue here that the shorter and more hazardous journey across the Skagerrak strait were used extensively during the warmer month of the year. This is further strengthened by the similarities in the archaeological material, particularly the earthen mounds, which cannot be found along the Swedish and Eastern Norwegian coast.

Allowing for the interaction between predicted boat performance data within multiple years of environmental data makes it possible to break down and analyse all the multitude of components that underpin sea travel.

The model developed to predict boat performance of the *Tilia Alsie*, can be adapted to any type of prehistoric or modern vessel provided its hull shape is digitally recorded, and there are sufficient recorded parameters of the vessel. This calls for the systematic recording of all types of prehistoric boat reconstructions [74], which includes the impact of waves on boat speed and crew endurance. Such information would allow for the direct comparison of results from different types of boats (prehistoric/modern whether, e.g., paddled/sailed), encountering the same multi-year environmental data, while travelling between points within the same

geographic time frame, or corresponding geographic time frames where coastal configuration has changed. This offers huge potential for future studies of seafaring in the past while also making it possible to reanalyse already existing studies of seafaring that have been based on other methods.

## Acknowledgments

We greatly thank the Viking Ship Museum in Roskilde for supplying the digital 3D hull data of the *Tilia Alsie* and Knut Valbjørn and Niels Peder Fenger, for helpful comments during the process of producing the polar diagrams and establishing reasonable limitations for these simulations.

## Author contributions

**Conceptualization:** Boel Bengtsson, Alvaro Montenegro.

**Data curation:** Boel Bengtsson, Matteo Tomasini, Ashely Green, Martyn Prince, Cecilia Lindhé.

**Formal analysis:** Boel Bengtsson, Alvaro Montenegro, Martyn Prince.

**Funding acquisition:** Johan Ling.

**Investigation:** Boel Bengtsson, Alvaro Montenegro, Ashely Green.

**Methodology:** Boel Bengtsson, Alvaro Montenegro, Matteo Tomasini, Ashely Green, Martyn Prince, Victor Wåhlstrand Skärström.

**Project administration:** Boel Bengtsson.

**Resources:** Cecilia Lindhé.

**Software:** Alvaro Montenegro, Matteo Tomasini, Victor Wåhlstrand Skärström.

**Supervision:** Boel Bengtsson, Alvaro Montenegro, Johan Ling.

**Validation:** Boel Bengtsson, Alvaro Montenegro.

**Visualization:** Boel Bengtsson, Alvaro Montenegro, Matteo Tomasini, Ashely Green, Knut Ivar Austvoll.

**Writing – original draft:** Boel Bengtsson, Matteo Tomasini, Martyn Prince, Knut Ivar Austvoll.

**Writing – review & editing:** Boel Bengtsson, Alvaro Montenegro, Ashely Green, Matteo Tomasini, Martyn Prince, Knut Ivar Austvoll, Johan Ling.

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
