## [Decision Letter · Decision Letter 0]

2 Dec 2024

PONE-D-24-35566Seafaring and navigation in the Nordic Bronze Age. Comparing direct open water crossings with sheltered coastal routes using an ocean voyage simulation tool.PLOS ONE

Dear Dr. Clark,

Thank you for submitting your manuscript to PLOS ONE. After careful consideration, we feel that it has merit but does not fully meet PLOS ONE’s publication criteria as it currently stands. Therefore, we invite you to submit a revised version of the manuscript that addresses the points raised during the review process.

**Please address all comments if the reviewer in detail before re-submission. **

We look forward to receiving your revised manuscript.

Kind regards,

Peter F. Biehl, PhD

Academic Editor

PLOS ONE

**Journal Requirements:**

This work is supported by Riksbankens Jubileumsfond under Grant M21-0018

We greatly thank the Viking Ship Museum in Roskilde for supplying the digital 3D hull data of the Tilia Alsie and Knut Valbjørn and Niels Peder Fenger, for helpful comments during the process of producing the polar diagrams and establishing reasonable limitations for these simulations. Sadly, Niels Peder Fenger passed away in the autumn 2023. This work was supported by Riksbankens Jubileumsfond under Grant M21-0018

This work is supported by Riksbankens Jubileumsfond under Grant M21-0018

4. In the online submission form, you indicated that your data is available only on request from a third party. Please note that your Data Availability Statement is currently missing the contact details for the third party, such as an email address or a link to where data requests can be made. Please update your statement with the missing information. 

5. We note that you have referenced "Scheen R. De norske, Holberg E. Klokkebegerkulturens and Klem PG." which has currently not yet been accepted for publication. Please respond by return e-mail with a copy of your updated manuscript to include to remove this from your References and amend this to state in the body of your manuscript: ("Scheen R. De norske, Holberg E. Klokkebegerkulturens and Klem PG." [Unpublished]) as detailed online in our guide for authors

http://journals.plos.org/plosone/s/submission-guidelines#loc-reference-style.   We can then upload this to your submission on your behalf.

6. We note that Figures 1, 2, 3, 4, 8, 10, 11 and 26 in your submission contain map images which may be copyrighted. All PLOS content is published under the Creative Commons Attribution License (CC BY 4.0), which means that the manuscript, images, and Supporting Information files will be freely available online, and any third party is permitted to access, download, copy, distribute, and use these materials in any way, even commercially, with proper attribution. For these reasons, we cannot publish previously copyrighted maps or satellite images created using proprietary data, such as Google software (Google Maps, Street View, and Earth). For more information, see our copyright guidelines: http://journals.plos.org/plosone/s/licenses-and-copyright.

We require you to either present written permission from the copyright holder to publish these figures specifically under the CC BY 4.0 license, or remove the figures from your submission:

a You may seek permission from the original copyright holder of Figures 1, 2, 3, 4, 8, 10, 11 and 26 to publish the content specifically under the CC BY 4.0 license.  

b If you are unable to obtain permission from the original copyright holder to publish these figures under the CC BY 4.0 license or if the copyright holder’s requirements are incompatible with the CC BY 4.0 license, please either i) remove the figure or ii) supply a replacement figure that complies with the CC BY 4.0 license. Please check copyright information on all replacement figures and update the figure caption with source information. If applicable, please specify in the figure caption text when a figure is similar but not identical to the original image and is therefore for illustrative purposes only.

7. We note that Figure 5 in your submission contain copyrighted images. All PLOS content is published under the Creative Commons Attribution License (CC BY 4.0), which means that the manuscript, images, and Supporting Information files will be freely available online, and any third party is permitted to access, download, copy, distribute, and use these materials in any way, even commercially, with proper attribution. For more information, see our copyright guidelines: http://journals.plos.org/plosone/s/licenses-and-copyright.

We require you to either present written permission from the copyright holder to publish these figures specifically under the CC BY 4.0 license, or remove the figures from your submission:

a You may seek permission from the original copyright holder of Figure 5 to publish the content specifically under the CC BY 4.0 license. 

8. Please upload a new copy of Figures 9 and 15 as the detail is not clear. Please follow the link for more information: "https://blogs.plos.org/plos/2019/06/looking-good-tips-for-creating-your-plos-figures-graphics/" "https://blogs.plos.org/plos/2019/06/looking-good-tips-for-creating-your-plos-figures-graphics/"

**Additional Editor Comments:**

Please address all comments if the reviewer in detail before resubmission.

Reviewers' comments:

Reviewer's Responses to Questions

**Comments to the Author**

1. Is the manuscript technically sound, and do the data support the conclusions?

Reviewer #1: Yes

2. Has the statistical analysis been performed appropriately and rigorously? 

Reviewer #1: Yes

3. Have the authors made all data underlying the findings in their manuscript fully available?

Reviewer #1: Yes

4. Is the manuscript presented in an intelligible fashion and written in standard English?

Reviewer #1: Yes

5. Review Comments to the Author

**Reviewer #1:**  Overall Summary:

This paper presents a fascinating application of computer simulations to model Bronze Age seafaring between Denmark and Norway. While the focus is on a specific region and time period, the methodology has significant potential for broader application by seafaring modelers, ethnographers, and experimental archaeologists working in similar contexts. It is a shame then that the authors make a claim that maritime activities “are most often illustrated merely by the addition of large arrows on a map.” Sadly, this overlooks a well-developed and sophisticated body of work already engaging in seafaring modelling. While the paper proposes novel approaches, it misses an opportunity to build on this existing corpus. Acknowledging and situating the study within the broader context of previous research would strengthen its contribution.

Recommendation:

The study presents a promising contribution to seafaring archaeology but requires revisions before publication. Addressing the methodological concerns, situating the work within existing literature, and improving the presentation of text and images would significantly enhance its impact and reach.

See attachments for full review and comments.

6. PLOS authors have the option to publish the peer review history of their article (what does this mean? ). If published, this will include your full peer review and any attached files.

**Do you want your identity to be public for this peer review?** For information about this choice, including consent withdrawal, please see our Privacy Policy .

Reviewer #1: No

---

## [Author Response · Author response to Decision Letter 1]

17 Feb 2025

Detailed responses to both editor and reviewer has been itemised and provided in the "response to reviewers" document uploaded.

Answers to Journal Requirements.

1.Endeavour to meet style requirements, including file naming.

2. There was no additional external funding received for this study. The statement has been amended to include ‘all’ the funding or sources of support etc. etc.

3.We have removed ‘ this work is supported by…’ from Acknowledgements and are happy with the phrase ‘ this work is supported by Riksbankens etc. etc.’

4. Email and name of contact will be included (see below in answer to reviewer)

5. The referenced unpublished material by Scheen, R, has been replaced with a reference to published material. This has been addressed in the manuscript.

6. Original copyright holder of Figures 1,2,3,4,8,10,11 and 26. These maps have been made by extrapolating data from SGU (https://www.sgu.se/produkter-och-tjanster/kartor/kartvisaren/jordkartvisare/strandforskjutningsmodell/), using a model by Påsse and Daniels 2015. The maps have been manipulated and created by Ashely Green and Boel Bengtsson, using ArcGIS Pro and Inkscape. Thus original copyright holders are Ashely and Bengtsson. In figure 3, two landscape drawings are included. Although in both cases the artists have been deceased for over 100 years, we have gained specific permission from Kristian Jensen, editor at Jysk Arkæologisk Selskab (this email will be uploaded for you to view). The other drawing here, showing Bulbjerg, is collected from an open source (https://commons.wikimedia.org/wiki/File:Bulbjerg-Skarreklit_JTH.jpg?uselang=en#Licensing), stating it is under no copyright restrictions for use.

7. Fhotos in figures 5 and 12 are taken by the lead author, Boel Bengtsson, and can be published under the CC BY 4.0 license.

8. All figures will be updated and resubmitted ensuring a high level of detail is retained using PACE .

---

## [Editor Report · Decision Letter 1]

25 Feb 2025

Seafaring and navigation in the Nordic Bronze Age. The application of an ocean voyage tool and boat performance data for comparing direct open water crossings with sheltered coastal routes.

PONE-D-24-35566R1

Dear Dr. Clark,

We’re pleased to inform you that your manuscript has been judged scientifically suitable for publication and will be formally accepted for publication once it meets all outstanding technical requirements.

Kind regards,

Peter F. Biehl, PhD

Academic Editor

PLOS ONE
---

## [Editor Report · Acceptance letter]

PONE-D-24-35566R1

PLOS ONE

Dear Dr. Clark,

I'm pleased to inform you that your manuscript has been deemed suitable for publication in PLOS ONE. Congratulations! Your manuscript is now being handed over to our production team.

Kind regards,

on behalf of

Dr. Peter F. Biehl

Academic Editor

PLOS ONE